# Task-Relevant Language-conditioned Segmentation for Robust Generalization in Reinforcement Learning

## Abstract

Humans possess a remarkable ability to filter out irrelevant sensory clutter, extracting only the information needed to anticipate and act within dynamic environments. Prior attempts to mitigate this through augmentation and masking strategies have improved robustness, but remain limited by computational overhead, weak semantic grounding, or instability in actor-critic training. Inspired by how language guides human perception, we introduce Task Relevant Language-conditioned Segmentation (TaLaS), a framework that leverages language-conditioned segmentation to impose semantic structure on visual observations. TaLaS employs a two-phase design where in the first phase, a lightweight masker is pre-trained on unaugmented, language-guided masks; in the second phase, a student masker is regularized with strong augmentations to enforce consistency. This yields a task-relevant feature extractor that improves policy stability and removes the need for online segmentation at inference time. To address the actor's deployment distribution shift, we employ asymmetric actor-critic training. TaLaS improves robustness to distractors and achieves strong performance under challenging visual shifts on RL-ViGen, with its most pronounced gains appearing in the video-hard settings where semantic distractor suppression is most critical. The benchmark includes challenging variants of the DeepMind Control Suite, Quadruped Locomotion and Dexterous Manipulation tasks. https://talas-rl.github.io/

## 1 INTRODUCTION

Visual Reinforcement Learning (RL) has made substantial progress in learning control policies directly from high-dimensional sensory observations, enabling agents to acquire complex behaviors without access to privileged state information (Levine et al., 2016; Laskin et al., 2020b). Despite these advances, the generalization ability of visual RL agents remains severely limited. In many commonly used benchmarks, observations are generated from simulators in which the visual scene is already strongly aligned with the task, often through carefully designed viewpoints, simplified backgrounds, or environment constructions in which most visible content is implicitly relevant to control (Ha & Schmidhuber, 2018; Hafner et al., 2018; Hansen et al., 2024). Under such conditions, agents can achieve strong in-distribution performance while relying on fragile correlations that do not persist under even modest visual changes. When evaluated under shifts in texture, illumination, background dynamics, camera perturbations, or the presence of distractors, performance often deteriorates sharply (Cobbe et al., 2019; Ma et al., 2025). The fundamental challenge lies in disentangling task-relevant signals from exogenous noise while ensuring that only meaningful information is incorporated in learned representations.

At its core, the problem is of selective representation learning. Pixel-based RL agents typically process the full visual field as a dense input tensor and must implicitly infer, from reward alone, which parts of the scene should be encoded and which should be ignored. This places a substantial burden on the optimization process. Because reward is sparse, delayed, and entangled with exploration, the agent is rarely given direct supervision about what in the image matters for control. As a result, learned representations can absorb noise factors that are predictive in the training distribution but spurious from the standpoint of task structure. Background patterns, distractor motion, lighting statistics, and incidental textures may all become embedded in the policy or critic representation if they correlate with reward during training.

Such entanglement reduces robustness under distribution shift and makes it difficult for the agent to transfer across environments that preserve task semantics but alter appearance. This stands in contrast to human perception, which is inherently structured and selective. Humans do not treat every visible pixel or region as equally relevant for action. Instead, perception is organized around semantically meaningful entities such as objects, object parts, spatial relations, and action-relevant affordances. Irrelevant segments are suppressed to prevent distraction, while attention is dynamically adapted as scene context changes (Nasr et al., 2008; Seidl et al., 2012). The visual system selectively prioritizes structure that is useful for the intended behavior. This mismatch underscores a fundamental gap between how agents and humans parse visual scenes.

Recent work has explored masking-based strategies that suppress irrelevant parts of the input to focus learning on task-relevant information, but these methods exhibit several limitations. (Grooten et al., 2024; Zhang et al., 2024; Huang et al., 2023). SGQN (Bertoin et al., 2022) uses Q-function saliency maps to mask observations and suppress distractors. Since supervision is value-derived and shifts with critic training, early saliency is noisy and hyperparameter-sensitive, leaving perception entangled with a partially learned critic and inheriting its biases. Focus-then-Decide (Chen et al. (2024), FTD) performs observation segmentation via the Segment Anything Model (Kirillov et al. (2023), SAM), followed by an attention selector that identifies objects for control. This requires segmentation at every step, leading to high computational overhead and limiting practicality in complex tasks. SAM-G (Wang et al., 2023) uses DINOv2–SAM correspondences with manual point annotations to generate masks that are applied at every step, resulting in high computational overhead and dependence on SAM for entire training and evaluation. Our method avoids per-task annotation and online segmentation by using language identifiers to guide SAM once, then distilling these priors into a lightweight masker trained under augmentations.

A central mechanism through which humans organize perception is language. Verbal descriptions and category labels are not merely communicative abstractions; they actively reshape perceptual processing by biasing attention toward relevant features and suppressing irrelevant alternatives (Lupyan, 2012; Lupyan & Ward, 2013). Language compresses the hypothesis space of perception. For example, when a person is instructed to *"Pick the red apple from the crowded shelf"*, the term *apple* constrains perception to certain shape, size and graspable structure. *Red* sharpens color filters, and *shelf* imposes a spatial prior, allowing irrelevant clutter or background motion to be suppressed. Language narrows the search space of perception into discrete categories, making raw sensory input tractable for decision-making (Yang & Zelinsky, 2009). This observation is particularly compelling for visual RL, where the main challenge is often not lack of information, but excess information without a mechanism for semantic filtering.

Yielding inspiration from these insights, we propose **Ta**sk-Relevant **La**nguage-conditioned **S**egmentation (TaLaS), a novel algorithm that integrates language-guided segmentation with mask distillation to improve generalization in visual reinforcement learning. To ensure computational efficiency, we adopt a two-phase training pipeline. In Phase I, natural language prompts are provided to a segmentation backbone (Cuttano et al. (2025), SAMWISE), which partitions clean observations into semantically grounded masks. These masks serve as supervisory signals for a lightweight convolutional masker, which is optimized to approximate the teacher outputs using trajectories generated under a random exploration policy $\pi_{\exp}$. This phase serves as an offline semantic distillation stage: the expensive segmentation model and language interface are used only to produce supervisory targets, while the student masker learns to infer task-relevant segmentation directly from raw images. In Phase II, after $T_{\exp}$ exploration steps, the segmentation backbone and language interface are removed, and the convolution masker is frozen to serve as a teacher. A second, noisy student masker is then introduced, which receives augmented observations (overlayed with additional image, Appendix A.3) and is trained via consistency regularization to reproduce the teacher's outputs on corresponding clean inputs (Figure 1). This stage performs a form of robustness transfer: semantic priors extracted from clean scenes are propagated into a compact model that remains stable under visual corruption. The resulting role-reversal distillation procedure yields a masker that is both semantically informed and resistant to nuisance variation, without requiring online foundation-model segmentation or language processing during RL optimization. To improve robustness to mask imperfections at deployment, TaLaS uses an asymmetric actor–critic update: during actor updates, the policy is conditioned on clean and augmented masked views, while the critic evaluating the action uses the clean view; during critic updates, the online critic is trained on clean and augmented masked views, while the bootstrap target

is computed from the clean view only. The learned masker is not an endpoint by itself, but a compact perceptual bottleneck that filters irrelevant visual content before downstream control. Under this view, the contribution of TaLaS is not a new segmentation model, but an RL training framework in which semantically grounded masking improves robustness, efficiency, and transfer across visually perturbed domains.

We evaluate TaLaS on **eleven** environments spanning three challenging benchmarks in RL ViGen (Yuan et al., 2023): the DeepMind Control Generalization Benchmark (Hansen & Wang, 2021), Quadruped Locomotion (Hansen & Wang, 2021) and Dexterous Manipulation (Rajeswaran et al., 2018). Across these domains, TaLaS is most effective in the challenging distractor-heavy settings, where semantic filtering directly targets the main source of distribution shift. In particular, under video-hard evaluation, TaLaS achieves strong gains over established visual RL baselines, while maintaining competitive performance in easier settings where distractor suppression is less critical.

## 2 Related Work

### 2.1 Generalization with Data Augmentation

RL agents exhibit significant generalization gaps, often suffering sharp performance drops in out-of-distribution settings as a result of overfitting and poor adaptability (Kirk et al., 2023; Jiang et al., 2023). To address this, data augmentation has been widely adopted in visual RL, with methods such as RAD (Laskin et al., 2020b), DrQ (Yarats et al., 2021), and DrQ-v2 (Yarats et al., 2022) using image transformations to improve robustness, whereas approaches like SVEA (Hansen et al., 2021), SODA (Hansen & Wang, 2021) and SADA (Almuzairee et al., 2024) regularize representations by mixing clean and augmented streams and enforcing constraints on encoder features. Information-theoretic methods pursue a similar goal of compression and invariance by penalizing redundant or noisy channels (You et al., 2022; Dave & Rueckert, 2024; Wang et al., 2024), but they too lack explicit semantic grounding, making it difficult to bifurcate which features should be suppressed or retained. Recent work further refines the augmentation toward more selective forms: SRM (Huang et al., 2022) suppresses high-frequency components to mitigate texture bias, but global filtering risks erasing motion-relevant detail. TLDA (Yuan et al., 2022a) perturbs pixels deemed irrelevant through a Lipschitz-based sensitivity analysis of the policy, yet this proxy depends on local smoothness assumptions and may misclassify low-salience but task-critical signals. CG2A (Liu et al., 2023) stabilizes training under multiple transforms by calibrating gradient magnitudes, yet its robustness is bounded by the augmentation set. While these approaches improve generalization, they still enforce invariances through input-level perturbations and lack explicit semantic structure, leaving generalization tied only to augmentations to preserve task-relevant information.

### 2.2 Masking Distractors in RL

Masking has emerged as a significant strategy to improve generalization in visual RL by suppressing distractors. SGQN (Bertoin et al., 2022) derives saliency masks from Q-function attribution to enforce value consistency, but supervision is critic-dependent and unstable early in training. MLR(Yu et al., 2022) employs random occlusion and latent reconstruction to promote dynamics-relevant encodings, yet remains object-agnostic, offering no guarantee of alignment with task semantics. InfoGating (Tomar et al., 2023) learns differentiable gates via inverse dynamics and Q-learning losses, but inherits biases from auxiliary objectives and risks occluding subtle cues. MaDi (Grooten et al., 2024) uses a reward-driven CNN masker to filter irrelevant pixels, achieving efficiency gains but suffering from sparse, delayed supervision. Recent methods leverage segmentation backbones: SAM-G (Wang et al., 2023) combines DINOv2–SAM correspondences with human prompts to generate task-relevant masks, while FTD (Chen et al., 2024) uses SAM and attention selectors. Both methods, however, use a segmentation model during training and inference, introducing substantial computational overhead and, in some cases, reliance on instance-level labels (Section 5.6.1). Overall, masking improves robustness, but existing methods remain limited by unstable signals (SGQN, InfoGating), lack of semantic grounding (MLR, MaDi), or reliance on expensive foundation-model segmentation (SAM-G, FTD).

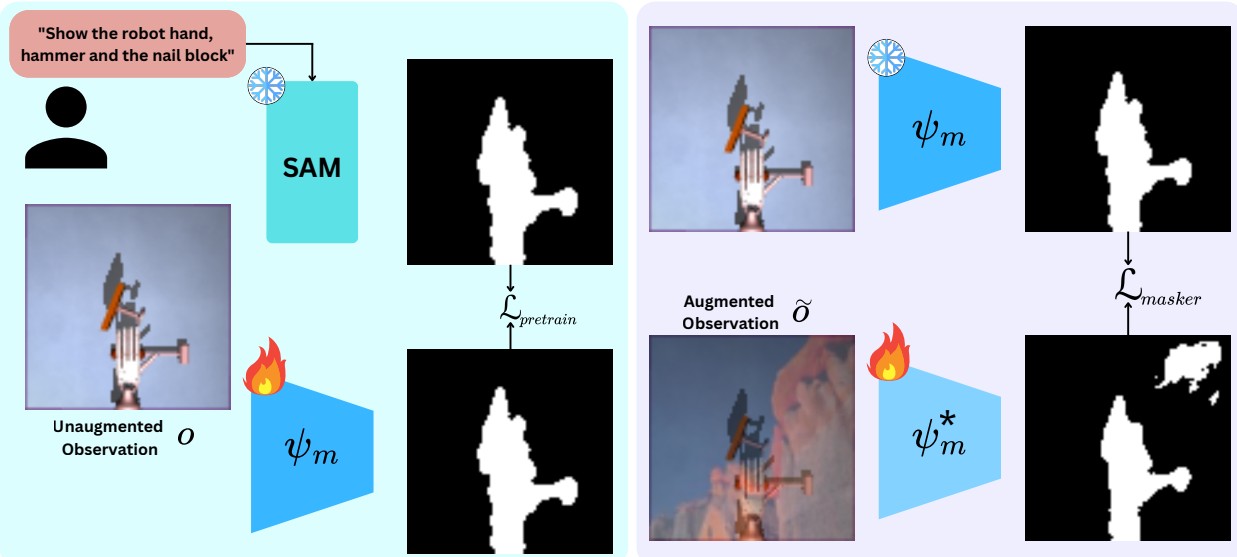

Figure 1: Depiction of the masker training phases of TaLaS. **Phase I (left): Language-conditioned distillation.** A text prompt drives a frozen segmentation model (SAMWISE) on unaugmented frames to produce task masks and a compact masker $\psi_m$ is pretrained by masked reconstruction to imitate these target masks. **Phase II (right): Augmentation-consistent student.** An unaugmented frame yields a target mask from the frozen teacher, while the corresponding augmented frame is fed to the student masker $\psi_m^\star$, which is trained through $\mathcal{L}_{\mathrm{masker}}$ to match the teacher mask. Actor-critic gradients are stopped at the masked observation. Modules marked with 🔥 are trainable, whereas modules marked with ❄ remain frozen.

## 3 Preliminaries

### 3.1 Visual Reinforcement Learning

Reinforcement learning (RL) solves sequential decision problems via interaction with an environment (Sutton & Barto, 2018). In visual settings, we model the task as a Partially Observable Markov Decision Process (POMDP) $\mathcal{M} = \langle \mathcal{S}, \mathcal{A}, \mathcal{O}, P, \Omega, R, \gamma \rangle$, where $P : \mathcal{S} \times \mathcal{A} \rightarrow \mathcal{P}(\mathcal{S})$ is the transition kernel, $\Omega : \mathcal{S} \rightarrow \mathcal{P}(\mathcal{O})$ the observation kernel, $R : \mathcal{S} \times \mathcal{A} \rightarrow \mathbb{R}$ the reward, and $\gamma \in [0, 1)$ the discount. To mitigate partial observability (Kaelbling et al., 1998), we use a $k$-frame stack $s_t = (o_t, o_{t-1}, \ldots, o_{t-k+1})$ as a practical proxy for the history, which enriches temporal context without full belief-state inference. The objective is to learn a policy $\pi_\phi$ maximizing discounted return $\mathbb{E}_{\tau \sim (\mathcal{M}, \pi_\phi)}[\sum_{t=0}^{\infty} \gamma^t R(s_t, a_t)]$, where trajectories $\tau$ is induced by the underlying dynamics of the POMDP.

### 3.2 Reinforcement Learning Backbone

We build on PIE-G (Yuan et al., 2022b), which couples a frozen pre-trained visual encoder with a lightweight off-policy actor–critic. A fixed image encoder $\phi_{enc}$ (e.g., ResNet (He et al., 2016)) projects masked observations to features $z_t = \phi_{enc}(o_t)$, where we use early-layer embeddings for better visual generalization and batch normalization to adapt to distribution shift. It is built on DrQ-v2 (Yarats et al., 2022), a DDPG (Lillicrap et al., 2015) style learner with twin critics and target networks. Given two critics $Q_{\theta_k}$ ($k \in \{1, 2\}$) and targets $Q_{\bar{\theta}_k}$, the critic minimizes

$$\mathcal{L}_Q(z) = \sum_{k=1}^{2} \mathbb{E}_{(o_t, a_t, r_t, o_{t+1}) \sim \mathcal{D}} \left[ (Q_{\theta_k}(z_t, a_t) - y_t)^2 \right], \qquad (1)$$

where the target is $y_t = r_t + \gamma \min_{k=1,2} Q_{\bar{\theta}_k}(z_{t+1}, a_{t+1})$, with $a_{t+1} = \pi_\phi(z_{t+1}) + \epsilon$, and $\epsilon \sim \text{clip}(\mathcal{N}(0, \sigma^2), -c, c)$. The actor is trained to maximize critic values via the objective $\mathcal{L}_\pi =$

$-\mathbb{E}\left[\min_{k=1,2} Q_{\theta_k}(z_t, a_t)\right]$. Following standard visual RL practice, we apply strong augmentation like overlay during training.

## 4 Task-Relevant Language-conditioned Masking

We address robust pixel-based control under distribution shift by enforcing task structure prior to policy learning. Training proceeds in two phases:

(i) A short distillation phase trains a compact masker to reconstruct masks generated by a frozen segmentation model.

(ii) A consistency phase trains a student masker on augmented samples to match the target masks from unaugmented samples, yielding augmentation-stable masks.

Control is learned via off-policy RL on masked inputs. Masking reduces distraction-induced variance, stabilizes value estimates under augmentation, and eliminates segmentation overhead at training and deployment.

### 4.1 Problem Statement

Our work addresses the challenge of generalization in visual reinforcement learning, where agents are trained in unaugmented or domain randomized environments, then evaluated in visually perturbed environments with unseen distractors. Given a set of POMDPs $\{\mathcal{M}_\nu : \nu \in \mathcal{V}\}$, sharing similar dynamics and reward structure, but varying in observation space $\mathcal{O}_\nu$ (due to distractions), our objective is to learn a policy $\pi$ that maintains consistent performance across this family of environments $\mathcal{M}$, despite shifts in the observation distribution in a zero-shot manner. Let $\mathcal{M}_{\text{train}}$ be the training environment and $\mathcal{M}_{\text{test}}$ a set of test environments. Following prior work (Kirk et al., 2023; Wang et al., 2024), the generalization gap of a policy $\pi$ is defined as $\nabla_{\text{gen}} := \eta_{\mathcal{M}_{\text{test}}}(\pi) - \eta_{\mathcal{M}_{\text{train}}}(\pi)$, where $\eta_{\mathcal{M}}(\pi)$ denotes the expected return of $\pi$ in environment $\mathcal{M}$. We assume that the policy behaves well on the training environment, as this metric remains susceptible to random policy (Jiang et al., 2023). Rather than formulating the problem as one of image segmentation, we view task-relevant masking as a structured perceptual prior for visual reinforcement learning. In TaLaS, language-conditioned segmentation is used only to provide semantic supervision for distillation, while the actual objective remains zero-shot policy generalization under visual distraction and appearance shift.

### 4.2 Mask Learning

To enforce semantic consistency across visually diverse environments, we construct task-relevant masked observations in two stages as described below. These masked observations serve as inputs to the downstream RL pipeline.

#### 4.2.1 Phase I: Language-Guided Mask Distillation

This first phase, as shown in Figure 1 (left), aims to distill spatial task structure into a compact teacher masker by learning to reconstruct masked images derived from a frozen segmentation model. For each environment, we define a single task-level natural language prompt $\ell \in \mathcal{L}$ at the beginning of training, before the mask-distillation stage and policy learning. The same prompt is then used throughout this entire phase to generate segmentation supervision. For a single environment, it is kept fixed across frames, episodes and random seeds, and is not adapted or tuned using policy performance or test-environment feedback. We have added the full list of task-level prompts in the Appendix A.1.2 (Table 3).

Given a clean input image $o \in \mathcal{O}^{H \times W \times C}$, and a natural language prompt $\ell \in \mathcal{L}$ describing the task semantics, a frozen language-conditioned segmenter $\Phi(o, \ell)$ produces a set of region proposals $\{m_k^+(o)\}_{k=1}^K \subset \{0,1\}^\Omega$. These proposals are then combined into a single task-relevant mask $m^+(o) \in [0,1]^\Omega$ via the logical OR operation $m^+(o) = \bigvee_{k=1}^K m_k^+(o)$. To amortize this computationally expensive segmentation process, we train a lightweight convolutional network masker $\psi_m$ to predict the mask $m(o) = \psi_m(o) \in [0,1]^\Omega$. The training

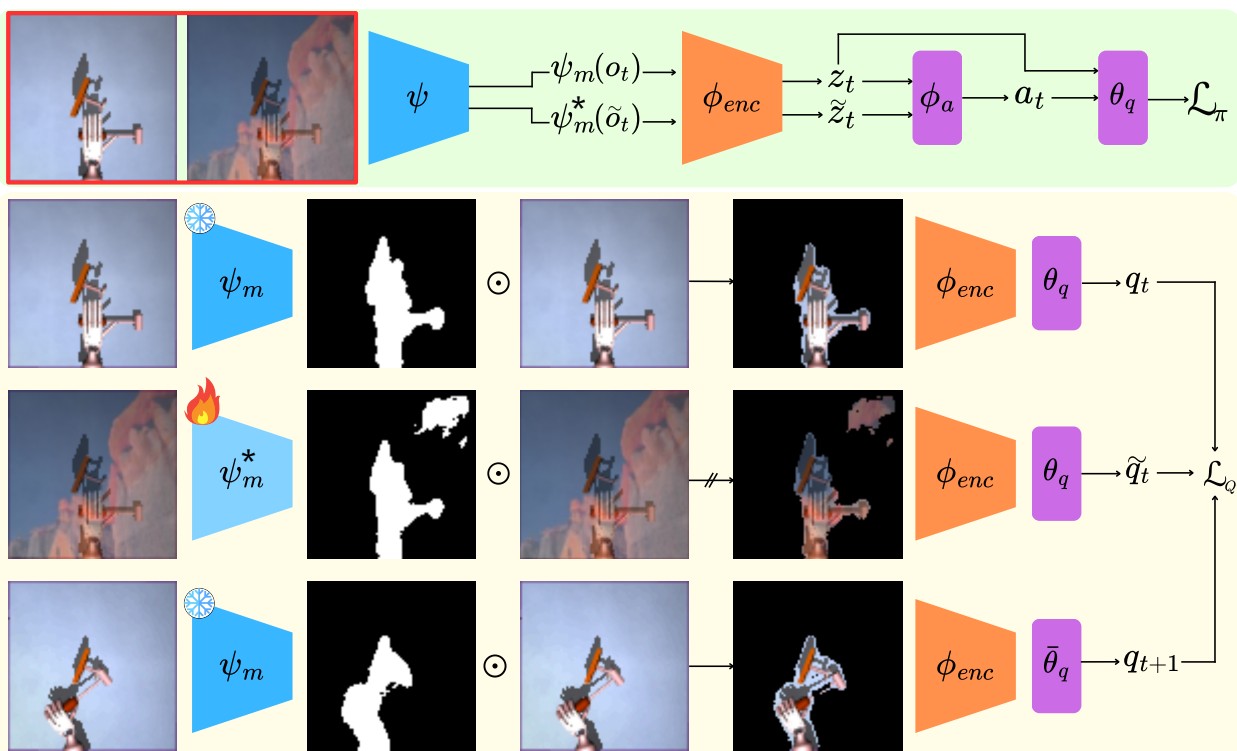

Figure 2: Integrating the masking strategy with the RL backbone for robust policy learning. **Actor.** The frozen masker $\psi_m$ filters clean inputs, while the student masker $\psi_m^\star$ filters augmented inputs. The encoder $\phi_{\text{enc}}$ maps both masked views to $z_t$ and $\tilde{z}_t$. The red rectangle denotes batch-wise concatenation $[z_t, \tilde{z}_t]_N$, where clean and augmented views are stacked as batch elements, not feature-concatenated. **Critic.** Critics $\theta_q$ use both clean and augmented masked views, while the target critic bootstraps only from the clean view. The stop-gradient sign blocks actor-critic gradients to $\psi_m^\star$, which is updated only by $\mathcal{L}_{\text{masker}}$. $\bar{\theta}_q$ denotes target critics; 🔥 and ❄ denote trainable and frozen modules respectively.

objective minimizes the binary cross-entropy between the ground-truth mask $m^+(o)$ produced by segmentation model and the predicted mask $m(o)$ by the convolution masker as $\mathcal{L}_{\text{pretrain}}(\psi) = \text{BCE}\left(m^+(o); m(o)\right)$. This formulation encourages the teacher to learn a spatial filter that preserves task-relevant content while suppressing irrelevant visual regions. These masks are collected during the initial exploration phase of the agent under random policy.

**But why use a random policy?** A random policy ensures diverse coverage of the observation space, exposing the segmenter to varied agent and object poses, crucial for learning generalizable masks. This avoids reliance on expert policies or co-training with control (online), keeping the segmentation phase modular and self-supervised. Once trained for $k$-iterations, $\psi_m$ serves as a teacher for the augmentation-consistent student masker trained in Phase II.

### 4.2.2 Phase II: Augmentation-Consistent Student Masker

For the second phase, as shown in Figure 1 (right), after pretraining the teacher masker $\psi_m$ on unaugmented images, we train a student masker $\psi_m^\star$ to produce task-relevant masks that are consistent under visual augmentations (Figure 1). The goal of this phase is to transfer the clean semantic prior learned by the teacher to visually corrupted observations, rather than to relearn task relevance from reward. The teacher masker $\psi_m$ has only seen unaugmented observations and therefore provides a stable clean-view semantic target. However, because the deployed agent may encounter augmented or visually shifted observations, directly using this clean teacher on corrupted inputs can introduce a distribution mismatch. Let $\tau \sim \mathcal{T}$ be a photometric transform that augments the image to $\tilde{o} = \tau(o)$. The teacher generates a mask $m(o) = \psi_m(o) \in [0, 1]^\Omega$ and to align this supervision with the augmented view, the student masker $\psi_m^\star$ then predicts a mask

$m^* = \psi_m^*(\tilde{o}) \in [0,1]^\Omega$ from the augmented input. The student is also trained with the teacher target via binary cross-entropy, $\mathcal{L}_{\text{masker}}(\psi_m^*) = \text{BCE}(m^*, m(o))$.

This teacher-student separation decouples semantic distillation from augmentation robustness. A single masker trained jointly on clean and augmented inputs would have to preserve clean semantic precision and become invariant to strong visual corruption using the same parameters. This simpler alternative entangles the semantic-supervision role and the augmentation-robustness role in a single network, which can weaken the clean semantic anchor under strong overlays. By keeping $\psi_m$ fixed and adapting only $\psi_m^*$, TaLaS preserves a stationary semantic target while learning a deployment-facing augmentation-robust masker. Importantly, the student masker is not optimized by the actor-critic objective. When the student is updated during the RL phase, it is updated only through this consistency loss against the frozen teacher, and gradients from the RL loss are stopped at the masked observation. For evaluation, the trained student masker $\psi_m^*$ is deployed as a frozen module generating task-consistent augmentation-invariant masks for downstream policy execution.

### 4.3 Reinforcement Learning with Masked Observations

To ensure robustness under distribution shift, we integrate the augmentation-consistent student masker $\psi_m^*$ within our RL framework (Figure 2). During RL, the teacher masker $\psi_m$ remains frozen. The student masker $\psi_m^*$ is used for augmented masked observations and, when updated, is updated only through the augmentation-consistency loss against the frozen teacher. Actor-critic gradients are stopped at the masked observation, so the RL objective updates only the actor and critic. At each timestep $t$, the agent receives a raw image observation $o_t$, which is processed by the frozen teacher masker $\psi_m$ to yield a binary mask $\psi_m(o_t) \in [0,1]^\Omega$. The clean masked observation is computed as the Hadamard product $o_t \odot \psi_m(o_t)$, and is passed through the encoder $\phi_{\text{enc}}$, yielding the clean masked representation $z_t = \phi_{\text{enc}}(o_t \odot \psi_m(o_t)) \in \mathbb{R}^d$. Similarly, given an augmented observation $\tilde{o}_t = \tau(o_t)$, the student masker $\psi_m^*$ produces an augmented masked observation $\tilde{o}_t \odot \psi_m^*(\tilde{o}_t)$, which is encoded as $\tilde{z}_t = \phi_{\text{enc}}(\tilde{o}_t \odot \psi_m^*(\tilde{o}_t)) \in \mathbb{R}^d$. Following SADA (Almuzairee et al., 2024), $z_t$ and $\tilde{z}_t$ are used as clean and augmented training streams, respectively, rather than as two simultaneous inputs to the actor. The actor remains a single-input policy $\pi_\phi(a_t \mid z_t)$. During actor updates, the same actor is applied to both $z_t$ and $\tilde{z}_t$, while the resulting actions are evaluated through the clean representation $z_t$. This exposes the policy to both clean and corrupted masked observations, while keeping value estimation anchored to the lower-variance clean stream. During critic updates, the critic is trained on both clean and augmented masked representations, but the bootstrap target is computed only from the clean next-state representation. Specifically, the target Q-value is $y_t = r_t + \gamma \min_{k=1,2} Q_{\bar{\theta}_k}(z_{t+1}, \pi_{\bar{\phi}}(z_{t+1}))$, where $z_{t+1} = \phi_{\text{enc}}(o_{t+1} \odot \psi_m(o_{t+1}))$. The critic loss is

$$\mathcal{L}_{\text{critic}} = \frac{1}{2} \sum_{k=1}^{2} \mathbb{E}_{(o_t, a_t, r_t, o_{t+1}) \sim \mathcal{D}} \left[ \left( Q_{\theta_k}(z_t, a_t) - y_t \right)^2 + \left( Q_{\theta_k}(\tilde{z}_t, a_t) - y_t \right)^2 \right].$$

This objective regresses both the clean masked representation $z_t$ and the augmented masked representation $\tilde{z}_t$ to the same clean-view bootstrap target. The target is computed from $z_{t+1}$, rather than an augmented next-state representation, to avoid injecting augmentation-induced variance into the temporal-difference target while still training the online critic on augmented masked inputs. The actor is updated using the asymmetric objective

$$\mathcal{L}_\pi(\mathcal{D}) = \mathbb{E}_{o_t \sim \mathcal{D}} \left[ -\min_{k=1,2} Q_{\theta_k}([z_t, z_t]_N, \pi_\phi([z_t, \tilde{z}_t]_N)) \right].$$

Here, $[\cdot, \cdot]_N$ denotes concatenation along the batch dimension, not concatenation along the feature dimension. Thus, $\pi_\phi([z_t, \tilde{z}_t]_N)$ means that the same single-input actor is applied batch-wise to both clean and augmented masked representations. Consequently, the clean counterpart of the augmented view is required only during the SADA-style training update and is not assumed to be available at deployment. At inference time, no clean counterpart is required; the deployed agent receives only the current observation and acts as $a_t = \pi_\phi(\phi_{\text{enc}}(o_t \odot \psi_m^*(o_t)))$.

Table 1: Performance on DMC Benchmark Environment in Video-Hard (VH) and Video-Easy (VE) settings. T: Tasks, CS: Cartpole Swingup, WW: Walker Walk, WS: Walker Stand, BiC: Ball in Cup Catch, FS: Finger Spin, CR: Cheetah Run.

| T(VE) | SAC | DrQ | DrQ-v2 | CURL | SVEA | SRM | PIEG | SGQN | MaDi | CNSN | TaLaS |
|---|---|---|---|---|---|---|---|---|---|---|---|
| CS | 398±60 | 485±105 | 267±41 | 404±67 | 782±27 | 724±75 | 482±51 | 717±35 | **848±6** | 353±40 | 531±65 |
| WW | 245±165 | 682±89 | 175±117 | 556±133 | 819±81 | 854±42 | 871±22 | 860±53 | 895±24 | **923±8** | 878±33 |
| WS | 389±131 | 873±83 | 560±48 | 852±75 | 961±8 | **963±57** | 957±12 | 955±9 | **967±3** | 956±9 | 961±13 |
| BiC | 192±157 | 318±157 | 871±106 | 316±119 | 871±106 | **924±35** | 910±37 | 761±171 | 807±144 | 892±43 | 856±48 |
| FS | 206±169 | 533±119 | 456±15 | 502±19 | 808±33 | **853±76** | 837±107 | 609±61 | 679±17 | 683±44 | 850±41 |
| CR | 87±21 | 102±30 | 64±22 | 104±24 | 249±20 | 257±21 | 287±20 | 269±33 | 294±25 | **347±34** | 219±31 |
| **Avg** | 253 | 499 | 457 | 456 | 757 | **763** | 724 | 697 | 748 | 692 | 716 |
| T(VH) | SAC | DrQ | DrQ-v2 | CURL | SVEA | SRM | PIEG | SGQN | MaDi | CNSN | TaLaS |
| CS | 158±17 | 138±9 | 130±3 | 114±15 | 393±45 | 475±75 | 323±24 | 488±18 | **619±24** | 309±19 | 395±35 |
| WW | 122±47 | 104±22 | 34±11 | 58±18 | 377±93 | 535±35 | 641±63 | 655±45 | 504±33 | 669±42 | **787±50** |
| WS | 231±57 | 289±49 | 151±13 | 45±5 | 834±46 | 863±57 | 852±56 | 851±24 | 824±287 | 856±38 | **940±22** |
| BiC | 101±37 | 100±40 | 97±27 | 115±33 | 403±174 | 566±135 | 773±74 | 782±57 | 758±135 | 721±7 | **864±86** |
| FS | 13±10 | 91±13 | 21±4 | 27±21 | 335±58 | 419±32 | 762±59 | 554±8 | 358±25 | 556±291 | **788±38** |
| CR | 10±5 | 32±13 | 23±5 | 21±7 | 105±37 | 115±24 | 154±17 | 144±34 | 170±14 | 162±23 | **198±22** |
| **Avg** | 106 | 126 | 76 | 63 | 408 | 496 | 584 | 579 | 539 | 545 | **662** |

# 5 Experiments

In this section, we present our experimental evaluations conducted on generalization benchmarks from RL-ViGen (Yuan et al., 2023). These benchmarks were selected as they encompass a wide range of environments: **(1)** DeepMind Control Generalization Benchmark (Hansen & Wang, 2021) for assessing continuous control under dynamic and reward variation; **(2)** Quadruped Locomotion (Zakka et al., 2022), which includes a Unitree quadruped robot, with complex balance and control demands; **(3)** Dexterous Manipulation (Rajeswaran et al., 2018), featuring multi-object interactions requiring precise strategies with sparse rewards.

## 5.1 Baseline Methods

We compare TaLaS against a comprehensive set of visual RL baselines designed to improve generalization: DrQ (Yarats et al., 2021), DrQ-v2 (Yarats et al., 2022), CURL (Laskin et al., 2020a), SVEA (Hansen et al., 2021), SRM (Huang et al., 2022), PIE-G (Yuan et al., 2022b), SGQN (Bertoin et al., 2022), MaDi (Grooten et al., 2024), CNSN(+PIEG) (Li et al., 2024), and VRL3 (Wang et al., 2022). These methods were chosen because they represent the most established approaches to generalization in visual RL, covering the main strategies explored in the literature, including data augmentation, pretrained encoders, saliency or masking, normalization, and learning from demonstrations. Across benchmarks, we compare against up to ten visual RL baselines. The exact set of baselines varies across benchmark suites due to differences in available implementations, compatible evaluation protocols, and reported results. Therefore, each table reports the strongest available baselines for the corresponding benchmark, rather than implying that every baseline is evaluated in every setting.

## 5.2 Zero-shot Evaluation.

We measure generalization without any fine-tuning on unseen environments spanning multiple distraction intensities. Specifically, we evaluate performance on the *video-easy* (10 background videos) and the more challenging *video-hard* (100 background videos without surface) configurations across all environments. Each seed undergoes evaluation over 100 episodes, corresponding to the designated noise levels.

Table 2: Performance on Unitree Walk and Unitree Stand task.

| Task (VE) | DrQ | DrQ-v2 | CURL | SVEA | SRM | PIEG | SGQN | TaLaS (ours) |
|---|---|---|---|---|---|---|---|---|
| Walk | 67±9 | 98±16 | 75±14 | 98±28 | 98±9 | 140±64 | 152±87 | **189±32** |
| Stand | 341±20 | 375±65 | 431±38 | **587±40** | 553±28 | 380±66 | 447±50 | 325±51 |
| **Average** | 204 | 236 | 253 | **343** | 326 | 260 | 332 | 257 |
| **Task (VH)** | **DrQ** | **DrQ-v2** | **CURL** | **SVEA** | **SRM** | **PIEG** | **SGQN** | **TaLaS (ours)** |
| Walk | 40±22 | 83±24 | 61±26 | 74±52 | 72±29 | 204±76 | 123±68 | **206±23** |
| Stand | 66±26 | 96±37 | 99±25 | 279±11 | **300±34** | 202±43 | 140±47 | **289±36** |
| **Average** | 53 | 89 | 80 | 177 | 186 | 203 | 131 | **247** |

## 5.3 Deepmind Control Suite

We evaluate our algorithm on the DMC-GB (Hansen & Wang, 2021) benchmark, spanning six tasks as shown in Table 1.

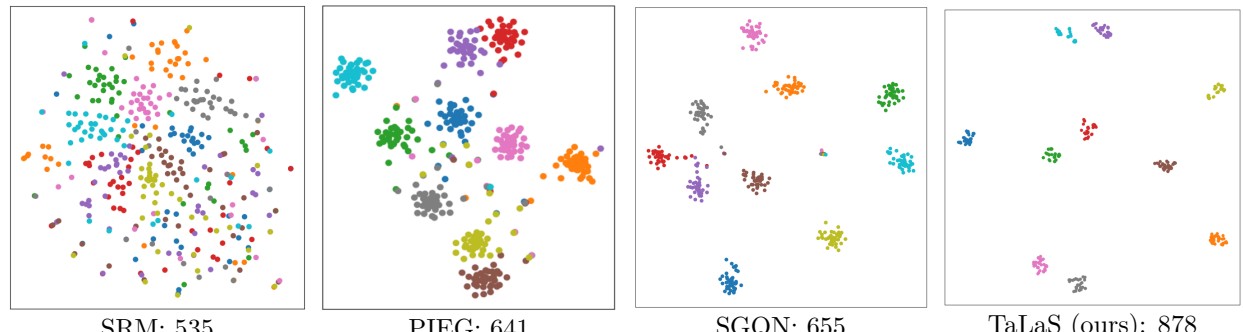

| SRM: 535 | PIEG: 641 | SGQN: 655 | TaLaS (ours): 878 |

Figure 3: t-SNE visualization of clustering results for TaLaS and three selected baseline methods.

### 5.3.1 Generalization Performance

We assess generalization across five seeds per task, reporting mean and standard deviation of episode returns. As shown in Table 1, TaLaS performs best in 7 of 12 environments. Its advantage is most pronounced in the challenging *video-hard* setting, outperforming all baselines with a **13.4%** gain over next-best method, PIEG, and **17.8%** over the average of the next-best four methods (PIEG, SGQN, CNSN, MaDi). In the *video-easy* setting, TaLaS remains competitive, trailing SRM by a modest **6%**. Although TaLaS achieves the best average performance in the video-hard setting, the gains are not uniform across all tasks. In particular, Cheetah Run remains an exception where methods such as CNSN and MaDi obtain higher returns. Nevertheless, the aggregate results show that TaLaS provides the largest improvements in distractor-heavy settings, where suppressing background variation is most critical.

### 5.3.2 Representations under Distractors

We evaluate domain invariance by visualizing encoder features via t-SNE (van der Maaten & Hinton, 2008). Ten states are augmented with 40 unseen backgrounds, and their embeddings are plotted with same-state points sharing a color. As shown in Figure 3, TaLaS yields the tightest clusters across background variations, indicating a strong ability to learn domain-invariant representations.

### 5.3.3 Robustness to Distraction Levels

We assess robustness by quantifying the performance drop from *video-easy* to *video-hard* settings using average returns. As shown in Table 1, most methods suffer significant degradation under elevated distraction. Even strong baselines like SRM and PIEG drop by 35% and 19% respectively. In contrast, TaLaS maintains

performance more consistently, with only a **7.5**% drop while still achieving high rewards. We further analyze Cheetah Run in Figure 8. This task depends on thin, fast-moving limbs, whose precise configuration is important for forward locomotion. Under video-hard distractors, the mask can partially attenuate low-contrast limb regions, which may explain why TaLaS is less competitive on this task despite strong aggregate video-hard performance.

### 5.4 Locomotion

For locomotion, we evaluate on Unitree Stand and Walk tasks from the Unitree Series (Zakka et al., 2022), using the same training and evaluation setup as in the DMC benchmark. Generalization is measured over three seeds per task by computing mean and standard deviation of returns. As shown in Table 2, TaLaS achieves the highest score on Unitree Walk in both video-easy and video-hard settings, outperforming all baselines. However, on Unitree Stand under video-easy, TaLaS yields a lower average score than most of the baselines. Nevertheless, under the more challenging video-hard configuration, TaLaS demonstrates robust performance across both tasks, achieving a **22**% gain in average return over the next-best method PIEG, highlighting its strong generalization to distractor-heavy environments.

### 5.5 Manipulation

In Adroit (Rajeswaran et al., 2018), we evaluate three tasks: Door, Hammer, and Pen, from a single view in RL-ViGen. The presence of numerous distractor objects requiring masking further increases the difficulty of this environment.

We evaluate generalization on five random seeds per task, reporting mean and standard deviation of returns. As shown in Figure 4, in the *video-easy* setting, TaLaS achieves an average improvement of **49**% over PIE-G, the strongest baseline, and **62**% over the average of the next three best methods (PIE-G, SRM, SGQN). In the more challenging *video-hard* setting, TaLaS still delivers substantial gains, surpassing PIE-G by **7**% and outperforming the average of PIE-G, SRM, and SGQN by **48**%. These results highlight the robustness of our masking strategy, which consistently yields better generalization under both easy and hard visual shifts. To make the aggregate comparison more robust, we additionally report interquartile mean (IQM) scores with stratified bootstrap confidence intervals in Appendix A.6.

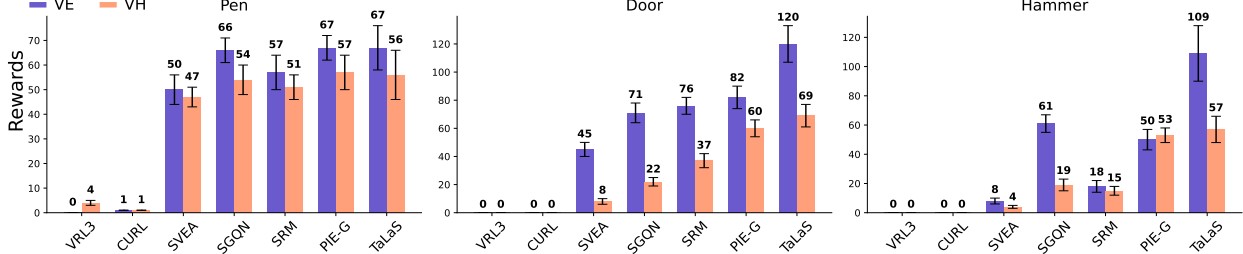

Figure 4: Performance on Adroit tasks (Pen, Door, Hammer) under *video-easy* (VE) and *video-hard* (VH) settings, averaged over 5 seeds. Error bars indicate standard deviation.

### 5.6 Ablation Study

#### 5.6.1 Wall Time

To assess the computational efficiency of our method, we compare the wall-clock times of TaLaS and a baseline that directly uses SAMWISE during both training and evaluation for Walker Walk task. For TaLaS, we report the actual wall time: $\approx 7.2$ seconds per 100 environment steps during training ($\approx 10$ hours), and $\approx 5$ seconds per evaluation episode. For the SAMWISE-based baseline, we report estimated latency, which amounts to $\approx 52$ seconds per 100 environment steps during training ($\approx 3$ days) and $\approx 150$ seconds per episode during evaluation, due to the cost of online segmentation. Because end-to-end runs with online SAMWISE segmentation were prohibitively expensive, we estimate its total wall time by measuring the per-

step segmentation latency and extrapolating it to the full training and evaluation horizon under the same hardware and environment configuration used for TaLaS. This demonstrates that TaLaS substantially reduces compute overhead while maintaining generalization, making it more practical for real-world deployments.

### 5.6.2 Actor Training

To quantify the effect of asymmetric training, we compare TaLaS with and without the proposed asymmetric actor–critic update. In the "without" setting, the actor is trained on only masked unaugmented observations. The critic training remains the same. Figure 5 shows the average reward in all the six DMC-GB environments, averaged over 5 seeds in both *video-easy* and *video-hard* settings. Asymmetric training yields substantial gains over both, easy and hard settings.

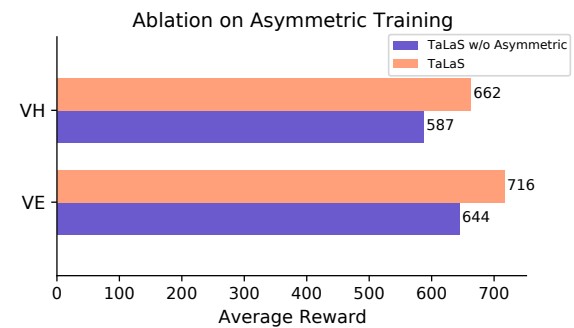

Figure 5: Ablation on asymmetric training in DMC.

### 5.6.3 Prompt sensitivity

To assess the effect of prompt specificity, we evaluate mask agreement under our default task-level prompts and a weaker coarse-prompt variant. For each task, we sample 10 random frames from the unaugmented setting and manually annotate task-relevant regions as human ground-truth masks. We then compute the IoU between the human annotation and the mask generated by either the default task-level prompt or the coarse prompt. The coarse prompts remove color cues and reduce semantic specificity. Averaged across all tasks, the default prompt set achieves a higher IoU of 0.6917, while the coarse prompt obtains 0.3890. For example, in Hammer, the default prompt specifies the yellow robot hand, hammer, and red nail block, whereas the coarse prompt only refers to the robot hand and object, leading to reduced mask agreement. Although a few tasks show task-specific deviations, the overall trend indicates that TaLaS is sensitive to prompt specificity, and that prompts naming the controllable body, task-relevant objects and their colors provide more accurate mask supervision. The full prompt variants, per-task IoU scores, human-annotation examples, and qualitative mask visualizations are provided in Appendix A.4.1.

### 5.6.4 Effect of Two-Stage Masker Distillation

To isolate the effect of the two-stage masker design, we compare full TaLaS against a single-stage masker variant on Walker Walk over five random seeds. In the single-stage variant, the masker is trained directly to predict SAMWISE masks without the teacher-student consistency stage. This variant achieves $842 \pm 61$ in the video-easy setting and $664 \pm 97$ in the video-hard setting, whereas full TaLaS achieves $878 \pm 33$ and $787 \pm 50$, respectively. This corresponds to a 4.3% improvement in video-easy and a 18.6% improvement in video-hard. These results indicate that the two-stage design provides a measurable benefit over directly training a single masker, with a larger gain under stronger visual distraction. This supports our claim that separating semantic distillation from augmentation-consistent student training improves the robustness of the agent under distractor-heavy settings.

## 6 Conclusion

We presented TaLaS, a two-stage framework for improving generalization in visual reinforcement learning by introducing semantic structure into the observation pipeline. Rather than requiring an RL agent to infer task relevance solely from reward, TaLaS uses language-conditioned segmentation to obtain task-aligned masks and distills these masks into a compact convolutional masker. In the first stage, a frozen segmenter provides semantically grounded supervision on clean observations; in the second stage, the distilled masker is regularized under strong augmentations to produce stable and augmentation-consistent masks. This design yields a lightweight perceptual bottleneck that suppresses nuisance variation while preserving action-relevant visual content, and removes the need for online segmentation during policy learning and deployment. To address the distribution shift induced by imperfect masks at test time, we combine this perceptual pipeline with an asymmetric actor–critic update. The actor is trained to operate under both clean and augmented

masked views, while the critic maintains stable value targets through clean-view bootstrapping. Empirically, this combination leads to the clearest gains under challenging distractor-heavy settings, where TaLaS consistently improves robustness and maintains strong performance without incurring the computational overhead of running a foundation segmentation model online. Taken together, these results support the broader view that robust visual control is not only a matter of stronger augmentation or larger encoders, but also of imposing the right semantic structure on what the agent is allowed to perceive.

At the same time, several limitations remain. First, the quality of the learned perceptual bottleneck depends on the quality of the language-conditioned supervision. If the task description is vague, incomplete, or semantically misaligned with the control objective, the resulting masks may omit relevant regions or preserve irrelevant ones. Second, the method depends on sufficient coverage during the initial exploration phase: if important objects, viewpoints, contact configurations, or interaction states are rarely observed early in training, the distilled masker may not learn to preserve them reliably once frozen. Finally, TaLaS is most naturally suited to settings in which the semantic identity of relevant objects remains stable across domains. When the distribution shift directly changes the appearance, structure, articulation, or visibility of the object of interest itself, segmentation-based filtering may become less reliable. These limitations point to several promising directions for future work. A first direction is to make the semantic interface more adaptive, for example through prompt refinement or prompt selection conditioned on the evolving task and state distribution. A second direction is to move beyond purely offline distillation toward iterative or online refinement of the masker during policy learning, so that the perceptual prior can be corrected when the agent encounters previously unseen but task-relevant structures. More broadly, it would be valuable to evaluate TaLaS in settings involving delayed object discovery, long-horizon interaction, stronger object-level distribution shifts, and real-world robotic data with dynamic backgrounds and sensor noise. Extending the framework to multimodal observations, where language-guided masking is combined with proprioceptive or tactile feedback, is another natural step toward more robust and deployment-ready embodied learning systems.

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

# A    Appendix

## A.1    Masker Implementation Details

### A.1.1    SAMWISE

We use SAMWISE, built atop the SAM2 (Ravi et al., 2025) backbone, to generate language-conditioned video masks. It integrates temporal and text cues via a Cross-Modal Temporal (CMT) adapter at intermediate layers and a Conditional Memory module to reduce tracking bias, all without fine-tuning SAM2 or relying on external VLMs. The added overhead is minimal (∼4.2–4.9M parameters), and the model operates in a streaming setup. A short task-specific prompt is used to obtain binary mask proposals.

### A.1.2    Natural language Prompts

For each environment, we define a fixed set of task-level prompts at the beginning of training, before the mask-distillation stage and before policy learning. The prompt is manually specified from the task description and names the controllable agent component and the task-relevant object or objects. The prompt is used only to query the frozen language-conditioned segmenter during Phase I. After the masks are distilled into the lightweight masker, the prompt interface and the segmentation model are removed from the reinforcement learning pipeline. Table 3 lists the prompts used for all environments in our experiments.

Table 3: Task-level prompts used for language-conditioned mask generation. Prompts are specified once at the beginning of training and kept fixed across frames, episodes, seeds, and evaluation domains.

| Benchmark | Task | Prompt(s) |
| --- | --- | --- |
| DMC-GB | Cartpole Swingup | "Show the orange cartpole" |
| DMC-GB | Walker Walk | "Show the orange walker" |
| DMC-GB | Walker Stand | "Show the orange walker" |
| DMC-GB | Ball in Cup Catch | "Show the orange cup", "Show the orange ball" |
| DMC-GB | Finger Spin | "Show the orange finger", "Show the orange rectangle" |
| DMC-GB | Cheetah Run | "Show the orange cheetah" |
| Unitree | Unitree Walk | "Show the black robot dog" |
| Unitree | Unitree Stand | "Show the black robot dog" |
| Adroit | Pen | "Show the yellow robot hand", "Show the pen" |
| Adroit | Door | "Show the yellow robot hand", "Show the orange door", "Show the door handle" |
| Adroit | Hammer | "Show the yellow robot hand", "Show the hammer", "Show the red nail block" |

### A.2  Learning Algorithm Implementation Details

#### A.2.1  RL Hyperparameters

We follow RL-ViGen (Yuan et al., 2023) defaults, using ResNet encoders (He et al., 2016) for PIE-G. For TaLaS, we set a masker learning rate of $10^{-4}$, $2\,\mathrm{k}$-step exploration phase ($T_{exp}$), $2\,\mathrm{k}$ masker pretraining steps, and batch sizes of 128 and 32 for RL and masker training, respectively. For SAMWISE (Cuttano et al., 2025), we retain its default setup, adjusting the input resolution to $90 \times 90$ for Hammer and Door in Adroit.

#### A.2.2  Masker Sigmoid

The masker is a compact CNN with three convolutional layers of channel dimensions $3 \rightarrow 64 \rightarrow 32 \rightarrow 1$, using kernel sizes $7 \times 7$, $5 \times 5$, and $3 \times 3$, respectively, with padding to maintain spatial resolution. Each layer is followed by Group Normalization and ReLU, except the final, which uses a temperature-controlled sigmoid $\max(0, \min(1, 1/(1+e^{-sx})))$ with slope $s = 5$ ($\tau = 0.2$), producing near-binary, differentiable masks that stabilize gradients.

### A.3  Augmentations

All baselines except SAC employ data augmentations. TaLaS uses SVEA-style overlay augmentation (Hansen et al., 2021) with random images from Places (Zhou et al., 2018) dataset: $\tilde{o}_t = \delta \cdot o_t + (1 - \delta)n$, where $n$ is a randomly selected background and $\delta = 0.5$. We use the same augmentation setting across all methods to ensure a controlled and fair comparison. The coefficient $\delta$ controls the strength of the visual corruption: larger values preserve more of the original observation, whereas smaller values increase the contribution of the distractor image. TaLaS is not architecturally tied to $\delta = 0.5$, since the student masker is trained to recover the clean teacher mask from corrupted observations. However, we do not claim invariance to arbitrary overlay strengths. Evaluating performance across different values of $\delta$ is a useful sensitivity analysis and is left for future work.

### A.4  Ablation Study

#### A.4.1  Prompt Ablation

To quantify prompt sensitivity, we sample 10 random frames per task from the unaugmented (original) setting and manually annotate task-relevant regions as human ground-truth masks. We then compute the intersection-over-union (IoU) between each prompt-induced mask and the corresponding human annotation. All masks are binarized before IoU computation.

Figure 6 illustrates the human-annotation protocol used to obtain the ground-truth masks. The prompt variants are listed in Table 4, the per-task IoU scores are reported in Table 5, and qualitative prompt-sensitivity examples are shown in Figure 7.

#### A.4.2  Qualitative Mask Analysis under Video-Hard Distractors

We provide additional qualitative mask examples to analyze the behavior of TaLaS under strong visual distractors. Figure 8 shows video-hard observations and the corresponding masked observations across representative tasks. The masked views preserve the controllable agent and task-relevant objects while suppressing background distractors.

### A.5  Comparison with FTD

To contextualize TaLaS against segmentation-assisted RL, we compare with FTD (Chen et al., 2024), one of the closest methods conceptually. The DMC tasks common to both papers are Cartpole Swingup, Finger Spin, Cheetah Run, and Walker Walk. We report the published FTD results on these shared tasks and compare them with TaLaS under the RL-ViGen video-hard setting. To further extend the comparison within our RL-ViGen setting, we additionally run FTD on Walker Stand over three random seeds and report

Table 4: Prompt variants used for prompt-sensitivity analysis. The default prompt corresponds to the prompt used in the main experiments. Coarse prompts remove color cues and reduce semantic specificity. IoU is computed against human-annotated task-relevant masks over 10 randomly sampled unaugmented frames per task.

| Task | Default prompt(s) | Coarse prompt(s) |
|---|---|---|
| Cartpole Swingup | "Show the orange cartpole" | "Show the cartpole" |
| Walker Walk | "Show the orange walker" | "Show the walker" |
| Walker Stand | "Show the orange walker" | "Show the walker" |
| Ball in Cup Catch | "Show the orange cup", "Show the orange ball" | "Show the cup and ball" |
| Finger Spin | "Show the orange finger", "Show the orange rectangle" | "Show the finger and object" |
| Cheetah Run | "Show the orange cheetah" | "Show the cheetah" |
| Unitree Walk | "Show the black robot dog" | "Show the dog" |
| Unitree Stand | "Show the black robot dog" | "Show the dog" |
| Pen | "Show the yellow robot hand", "Show the pen" | "Show the hand and object" |
| Door | "Show the yellow robot hand", "Show the orange door", "Show the door handle" | "Show the hand and door" |
| Hammer | "Show the yellow robot hand", "Show the hammer", "Show the red nail block" | "Show the hand and object" |

Table 5: Per-task prompt sensitivity under unaugmented observations. IoU is computed against human-annotated task-relevant masks using 10 randomly sampled frames per task. The default prompt corresponds to the prompt used in the main experiments.

| Task | Default prompt IoU | Coarse prompt IoU |
|---|---|---|
| Cartpole Swingup | 0.8394 | 0.0209 |
| Walker Walk | 0.6399 | 0.1799 |
| Walker Stand | 0.8064 | 0.6694 |
| Ball in Cup Catch | 0.6059 | 0.6173 |
| Finger Spin | 0.7808 | 0.6856 |
| Cheetah Run | 0.8840 | 0.6853 |
| Unitree Walk | 0.7270 | 0.4092 |
| Unitree Stand | 0.4039 | 0.4315 |
| Pen | 0.5008 | 0.1949 |
| Door | 0.5897 | 0.0016 |
| Hammer | 0.8307 | 0.3835 |
| Average | 0.6917 | 0.3890 |

the result in Table 6. Running FTD requires online segmentation during policy learning and evaluation, making full five-seed reproduction across all DMC tasks computationally prohibitive within the rebuttal window.

As shown in Table 6, TaLaS outperforms FTD on Cartpole Swingup, Finger Spin, and Walker Walk, while FTD performs better on Cheetah Run. This supports the overall trend that TaLaS is particularly effective under distractor-heavy settings, while also revealing Cheetah Run as a challenging case where thin, fast-moving body parts may make motion-relevant cues more sensitive to masking. The additional Walker Stand experiment further evaluates FTD under our RL-ViGen setting. We complement this performance com-

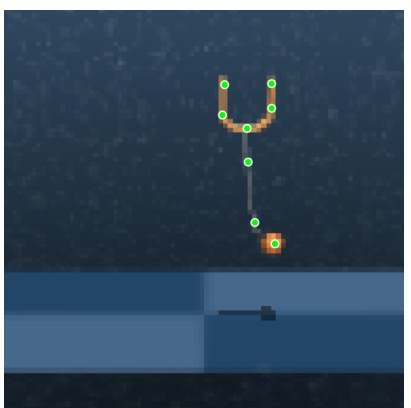 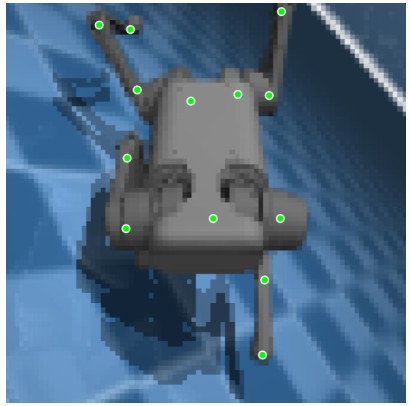 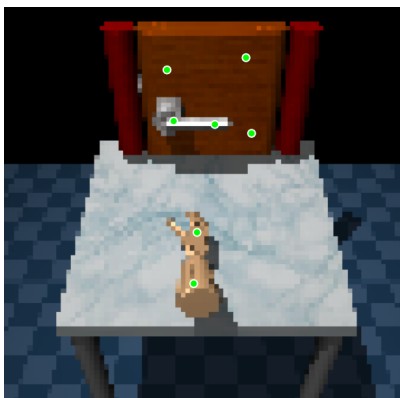

DMC: Ball-in-Cup Catch      Unitree Stand      Adroit: Door

Figure 6: Human-annotation for prompt-sensitivity evaluation. We manually annotate task-relevant regions on 10 randomly sampled unaugmented frames per task and use these annotations as ground-truth masks for IoU computation. The examples show annotations from Ball-in-Cup Catch, Unitree Stand, and Hammer.

parison with the wall-time analysis in Section 5.6.1, which shows the computational advantage of replacing online segmentation with a distilled lightweight masker.

Table 6: Comparison with FTD on DMC tasks. For tasks common to both papers, we report the published FTD results. Since Walker Stand is not reported in the original FTD paper, we additionally run FTD on Walker Stand over three random seeds and report the mean and standard deviation. TaLaS results correspond to the RL-ViGen video-hard setting.

| Task | FTD | TaLaS, video-hard |
|------|-----|-------------------|
| Cartpole Swingup | $207 \pm 26$ | $395 \pm 35$ |
| Finger Spin | $591 \pm 146$ | $788 \pm 38$ |
| Cheetah Run | $229 \pm 43$ | $198 \pm 22$ |
| Walker Walk | $395 \pm 49$ | $787 \pm 50$ |
| Walker Stand | $864 \pm 79$ | $940 \pm 22$ |

## A.6 Aggregate Adroit Statistics

To provide a robust aggregate comparison across Adroit tasks, we report interquartile mean (IQM) scores with 95% stratified bootstrap confidence intervals. Since Adroit tasks have different reward scales, returns are first normalized within each task and evaluation setting across methods and seeds. We then compute IQM over the pooled normalized scores across Pen, Door, and Hammer. The aggregate values are reported in Table 7 and visualized in Figure 9.

Table 7: Aggregate Adroit performance using IQM with 95% stratified bootstrap confidence intervals. Scores are normalized per task and evaluation setting before aggregation.

| Method | VE IQM 95% CI | VH IQM 95% CI |
|--------|---------------|---------------|
| VRL3 | 0.000 [0.000, 0.000] | 0.005 [0.000, 0.050] |
| CURL | 0.001 [0.000, 0.018] | 0.000 [0.000, 0.010] |
| SVEA | 0.346 [0.057, 0.647] | 0.213 [0.059, 0.680] |
| SGQN | 0.601 [0.478, 0.880] | 0.431 [0.244, 0.821] |
| SRM | 0.533 [0.132, 0.745] | 0.499 [0.209, 0.773] |
| PIE-G | 0.643 [0.390, 0.888] | 0.824 [0.726, 0.900] |
| TaLaS | 0.888 [0.816, 0.950] | 0.866 [0.813, 0.920] |

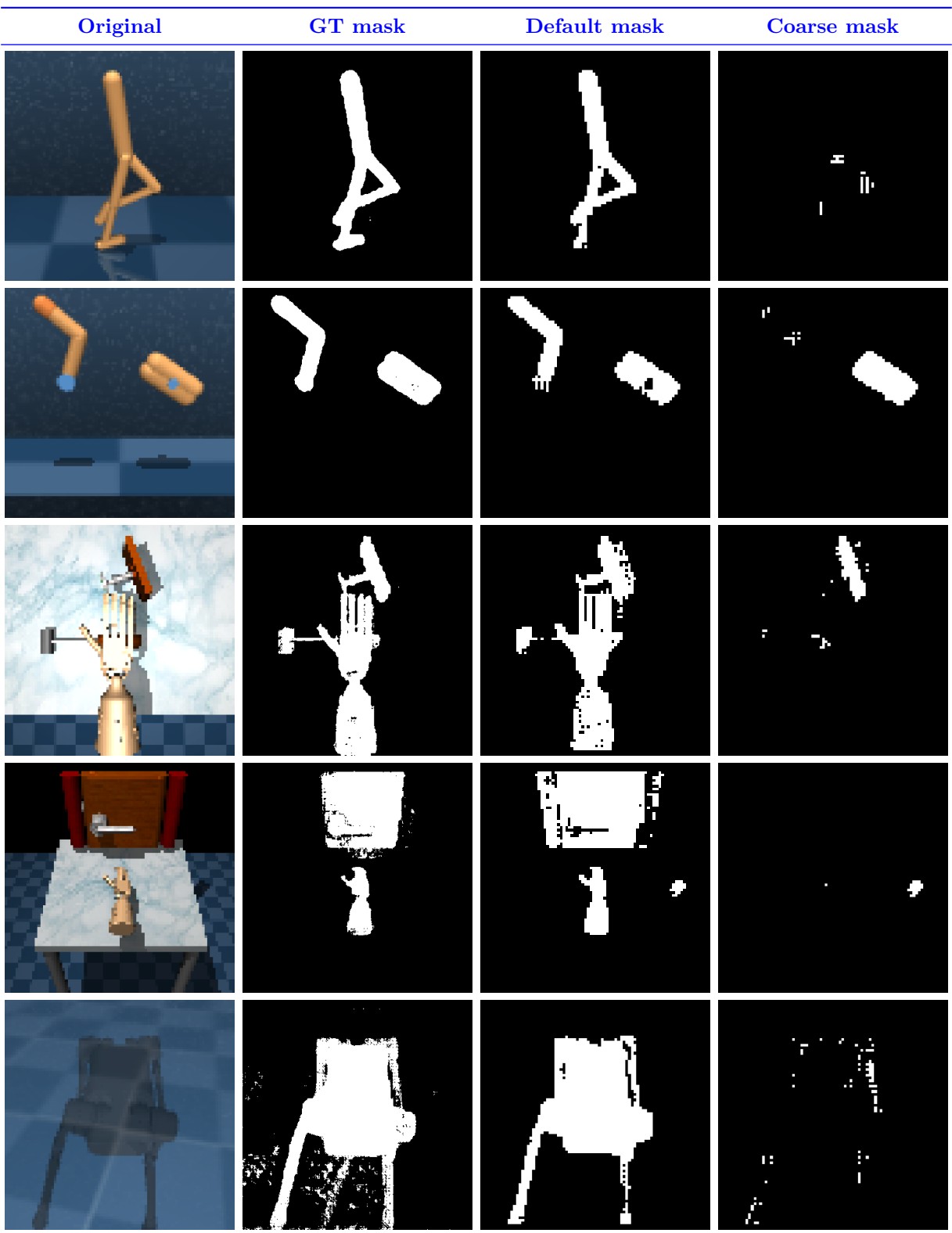

Figure 7: Qualitative prompt-sensitivity examples on unaugmented observations. Each row corresponds to one representative task. Columns show the original observation, human-annotated ground-truth mask, default-prompt mask, and coarse-prompt mask. The rows correspond to: first row, Walker Walk; second row, Finger Spin; third row, Hammer; fourth row, Door; and fifth row, Unitree Walk.

| Video-hard observation | Masked observation |
|---|---|

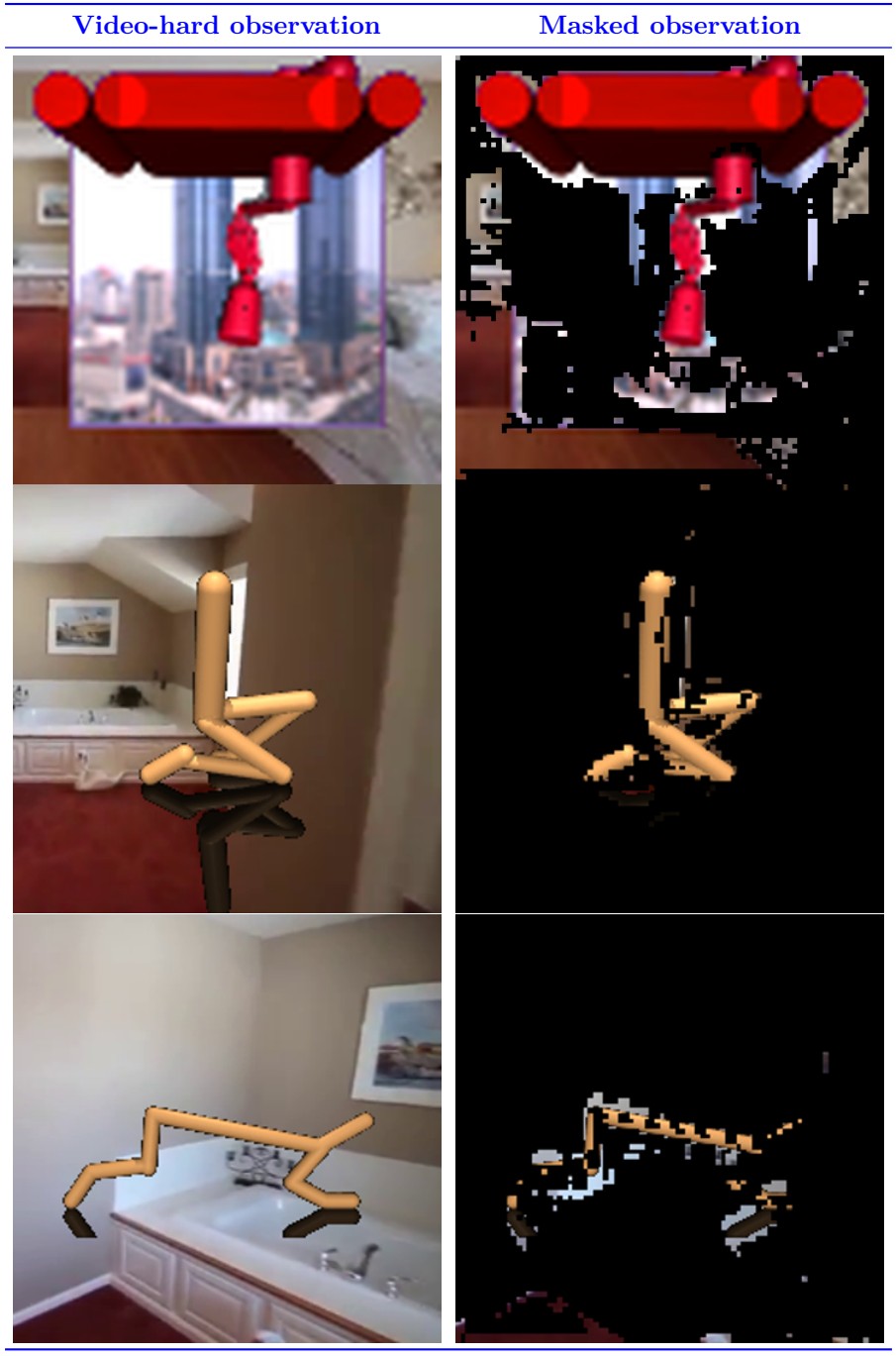

Figure 8: Qualitative mask examples under video-hard distractors. The rows correspond to: first row, *Door Opening*; second row, *Walker Walk*; third row, *Cheetah Run*. Columns show the original video-hard observation and the corresponding masked observation produced by TaLaS.

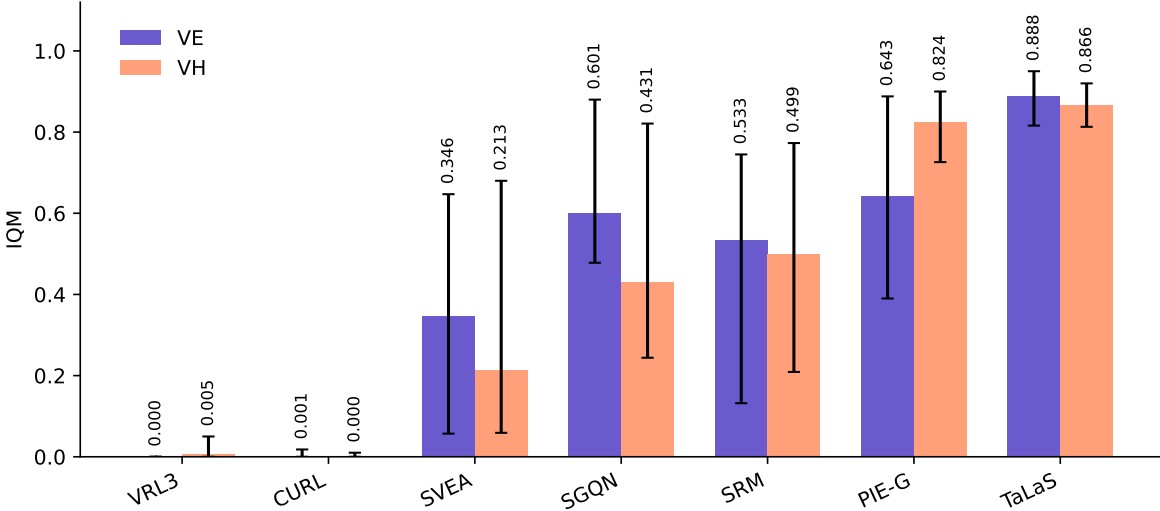

Figure 9: Aggregate Adroit performance measured using interquartile mean (IQM). Error bars denote 95% stratified bootstrap confidence intervals. Scores are normalized per task and evaluation setting before aggregation.

