# OpenReview forum: "Task-Relevant Language-conditioned Segmentation for Robust Generalization in Reinforcement Learning"
_TMLR — Decision pending for TMLR_

### Review · Reviewer_83Au · 2026-05-21

**Summary Of Contributions:**

This manuscript studies the zero-shot generalization problem in visual reinforcement learning. The aim is to learn a strategy to focus on task related regions without depending on visual cues from the training environment. Task-Relevant Language-conditioned Segmentation (TaLaS) is proposed to generate task related masks. These masks are then distilled into a lightweight masker. Segmentation serves as a tool to improve the robustness of visual RL policies under visual perturbations.

**Audience:**

Yes

**Audience Explanation:**

Yes. At least some of the TMLR audience would likely be interested in this paper. The paper addresses zero-shot generalization problems under visual distractions. The adoption of language-conditioned segmentation as a task relevant perceptual prior is also relevant to current interests in foundation models and semantic representation learning. The level of interest may depend on whether readers value applied robustness in visual RL because the paper is more focused on integrating existing segmentation models into RL.

**Broader Impact Concerns:**

None foreseen.

**Claims And Evidence:**

No

**Claims Explanation:**

The main claim of the paper is that TaLaS can improve zero-shot generalization in visual RL under visual distractors. This claim is supported by experiments including DeepMind Control, Quadruped Locomotion, and Dexterous manipulation tasks. In the more challenging video-hard setting, TaLaS often outperforms strong baselines. The t-SNE visualization also shows that the learned representations are less sensitive to background changes.

However, some claims would be stronger with additional ablation analysis. The paper mainly ablates the asymmetric actor-critic training. It will be better if the authors can provide ablation analysis to isolate the contributions of language-conditioned segmentation, the distillation, the student masker, etc. Besides, the manuscript does not provide evidence about how manual prompt design affects performance.

**Requested Changes:**

1. The current ablation mainly studies the asymmetric actor-critic design. But TaLaS has several components, it would be useful to isolate their effects.

For example, the authors should report results:
a. without Phase II student training, only using phase I teacher (to prove if augmentation consistency of student is necessary),
b. Only use a single stage masker to learn SAMWISE mask without student-teacher consistency (to see if two-stage distillation is better than one-stage masker training).

This will make it clearer which part of the method contributes most to the final performance.

2. The method depends on task-specific language prompts, but the paper does not fully discuss how these prompts are chosen. Please add analysis, if possible, to show the performance changes when the prompt is less specific or contains imperfect task descriptions, etc.

3. The manuscript should include more analysis about masks. For example, the authors could show mask examples across several tasks and under video-hard distractors. It would also help to report whether mask quality correlates with RL performance. Visualization of SAMWISE masker, Phase I and Phase II masker can be added.

4. Some experiments use only three seeds. Visual RL can have high variance, so more seeds or confidence intervals would make the results more convincing.

5. Some parts of the method are hard to follow, especially the role switch between the teacher masker and the student masker, and the asymmetric actor-critic update.

---

> ### Author Response · Authors · 2026-06-06
> **Response to Reviewer 83Au**
>
> We thank the reviewer for the careful and constructive evaluation of our submission. We appreciate the recognition of TaLaS’s relevance to zero-shot visual RL generalization, particularly under video-hard distractors. The comments helped us strengthen both the empirical analysis and the method presentation. We have revised the manuscript accordingly and address the reviewer’s points below. The revision in the manuscript is highlighted in blue.
>
> > *The current ablation mainly studies the asymmetric actor-critic design. But TaLaS has several components, it would be useful to isolate their effects.
> For example, the authors should report results: a. without Phase II student training, only using phase I teacher (to prove if augmentation consistency of student is necessary), b. Only use a single stage masker to learn SAMWISE mask without student-teacher consistency (to see if two-stage distillation is better than one-stage masker training). This will make it clearer which part of the method contributes most to the final performance.*
>
> We thank the reviewer for this helpful suggestion. We have added new results in Section 5.6.4, titled “Effect of Two-Stage Masker Distillation,” to better isolate the contribution of the masker-distillation design.
>
> In this experiment, we compare full TaLaS against a single-stage masker variant on Walker Walk over five random seeds. In the single-stage variant, the masker is trained directly to predict SAMWISE masks, without the teacher-student consistency stage used in TaLaS. The single-stage variant achieves $842 \pm 61$ in the video-easy setting and $664 \pm 97$ in the video-hard setting, whereas full TaLaS achieves $878 \pm 33$ and $787 \pm 50$, respectively. This corresponds to a $4.3\\%$ improvement in video-easy and a $18.6\\%$ improvement in video-hard.
>
> These results show that the two-stage design provides a measurable benefit over directly training a single masker, with the larger gain appearing under stronger visual distraction. This supports our claim that separating semantic distillation from augmentation-consistent student training improves the robustness of the agent.
>
>
> > *The method depends on task-specific language prompts, but the paper does not fully discuss how these prompts are chosen. Please add analysis, if possible, to show the performance changes when the prompt is less specific or contains imperfect task descriptions, etc.*
>
> We thank the reviewer for raising the issue of prompt sensitivity. We agree that, since TaLaS uses language-conditioned segmentation to obtain mask supervision, the effect of prompt specificity should be analyzed.
>
> We have revised the manuscript to address this in both the main paper and the appendix. First, in Section 4.2.1 and Appendix A.1.2, we now clarify that prompts are manually specified once from the task description, kept fixed across frames, episodes, seeds, and evaluation domains, and used only during the segmentation-supervision stage. We also provide the full list of task-level prompts in Appendix A.1.2.
>
> Second, we added a new prompt-sensitivity analysis in Section 5.6.3. For every prompt variant, we directly evaluate the effect of prompt specificity at the mask-supervision level. For each task, we sample 10 random unaugmented frames, manually annotate task-relevant regions, and compute IoU between the human annotation and masks generated by either the default task-level prompt or a weaker coarse prompt. The default prompts achieve higher average IoU than the coarse prompts, indicating that prompts naming the controllable body, task-relevant objects, and relevant visual attributes provide more accurate segmentation supervision. The full prompt variants, per-task IoU scores, human-annotation examples, and qualitative mask visualizations are provided in Appendix A.4.1.
>
>
> > *The manuscript should include more analysis about masks. For example, the authors could show mask examples across several tasks and under video-hard distractors. It would also help to report whether mask quality correlates with RL performance. Visualization of SAMWISE masker, Phase I and Phase II masker can be added.*
>
> We thank the reviewer for this suggestion. We have added additional qualitative mask analysis in the appendix/supplementary material. Specifically, we include examples under the video-hard setting, where each example shows the original observation together with the corresponding masked observation produced by TaLaS. These visualizations are provided across representative tasks and illustrate how the learned masker suppresses distractor-heavy background content while preserving the controllable body and task-relevant objects.
>
> We also include a supplementary Door opening video to show the temporal behavior of the masks during manipulation. These additions complement the prompt-sensitivity analysis and human-annotation-based IoU results already reported in Section 5.6.3 and Appendix A.4.1.

---

> ### Author Response · Authors · 2026-06-06
> **Response to Reviewer 83Au-II**
>
> > *Some experiments use only three seeds. Visual RL can have high variance, so more seeds or confidence intervals would make the results more convincing.*
>
> We are currently extending the experiments that were previously reported with three seeds to five seeds, in line with the reviewer’s suggestion. Once these additional runs are complete, we will update the revised manuscript with the new mean and standard deviation values, included together with the additional changes suggested by Reviewer cz8Y.
>
>
> > *Some parts of the method are hard to follow, especially the role switch between the teacher masker and the student masker, and the asymmetric actor-critic update.*
>
> We thank the reviewer for noting that the role switch between the teacher and student maskers and the asymmetric actor-critic update were difficult to follow. We have revised Section 4.2.2, Section 4.3, and Figure 2 to make these components explicit.
> In Section 4.2.2, we now clarify that the Phase I masker $\psi_m$ is trained from SAMWISE masks on clean observations and then frozen as a teacher. The Phase II student masker $\psi_m^\star$ receives augmented observations and is trained to match the frozen teacher through the augmentation-consistency loss. We also explicitly state that the student masker is not optimized by the actor-critic objective, and that actor-critic gradients are stopped at the masked observation.
>
> In Section 4.3, we rewrote the asymmetric actor-critic update to distinguish clean and augmented training streams. The actor remains a single-input policy: the clean and augmented representations $z_t$ and $\tilde{z}_t$ are concatenated along the batch dimension and passed through the same actor, rather than being used as simultaneous inputs to a two-input policy. We also added the clean-view bootstrap target, the explicit critic loss over both clean and augmented masked representations, and the inference-time policy path, which uses only the current observation and the trained student masker.
>
> Finally, Figure 2 and its caption were updated to show the batch-wise concatenation with a red rectangle and the stop-gradient marker indicating that actor-critic losses do not update $\psi_m^\star$.

---

> ### Author Response · Authors · 2026-06-08
> **Additional Response to Reviewer 83Au**
>
> > *Some experiments use only three seeds. Visual RL can have high variance, so more seeds or confidence intervals would make the results more convincing.*
>
> We thank the reviewer for raising this important point. We agree that visual RL can exhibit high seed-level variance and that additional seeds make the empirical comparison more reliable. In the latest revision, we have updated the Adroit manipulation experiments from three to five random seeds and revised Section 5.5 and Figure 4 accordingly. The results are now reported as mean and standard deviation over five seeds for Pen, Door, and Hammer.
>
> In addition, following the **cz8Y** reviewer’s suggestion, we added aggregate IQM statistics with (95%) stratified bootstrap confidence intervals in Appendix A.6. These aggregate statistics provide a more robust summary of the Adroit results across tasks and seeds.

---

### Review · Reviewer_AcQJ · 2026-05-22

**Summary Of Contributions:**

This paper presents a method for improving generalization in visual RL under visual distractors. It uses language-conditioned segmenter to segment task-relevant regions, and trains two lightweight maskers to predict these masks from unaugmented and augmented inputs, respectively. The maskers are then used to generate masked inputs during RL training and inference, to filter out noisy background. This is more efficient than using language-conditioned segmenters directly during RL. An asymmetric training method is also proposed to avoid overfitting to idealized inputs. Experimental results show good performance especially on hard benchmarks and benefit in computational efficiency.

**Audience:**

Yes

**Audience Explanation:**

Researchers in robotics and visual RL may find the proposed idea interesting.

**Claims And Evidence:**

No

**Claims Explanation:**

Although the experimental results are quite extensive, many key components of the method are unclear.

- It is unclear how the natural language prompt used by the language-conditioned segmenter is obtained. Is it manually written for each task and input, or generated automatically from the task description in some way? The prompt must be accurate in describing what is relevant to the task, so it deserves more explanation.

- The actor input is confusing as it includes representations of both the clean (unaugmented) and augmented images ($z_t$ and $\tilde{z}_t$). Why is it designed this way? At inference time, how is the clean representation available when the agent only observes inputs with noisy background?

- The asymmetric training strategy can be better explained. The sentence "During critic updates, the online critic is optimized on clean and augmented masked views, whereas the bootstrap target is computed from the clean view only." is somewhat confusing. Can the authors provide explicit equations for the training loss for the critic?

- It is unclear why the student masker is trainable also during RL. The student masker has been trained in the Augmentation-Consistent Masker Phase. Keeping it trainable during RL introduces undesirable variation and instability. Why is this needed or preferred?

- The need for two separate maskers is not fully justified. It is unclear why a single masker could not be trained on both unaugmented and augmented inputs. Can the authors compare with this simpler alternative or justify the original design?

**Requested Changes:**

Besides addressing the weaknesses in the "Claims" section, below are additional comments.

- The augmentation setup uses only one overlay strength ($\delta=0.5$). The authors should discuss whether the method could generalize to other values or report the results in those cases.


- Grammar: "our objective is to learn a policy $\pi$ consistent policy performance across this family of environments..."
- Grammar: "Both, however, utilizes segmentation model..., adding to substantial computational and overhead reliance on instance-level labels."

- POMDP should be defined.

---

> ### Author Response · Authors · 2026-05-27
> **Response to Reviewer AcQJ**
>
> We thank the reviewer for the careful and constructive evaluation of our submission. We appreciate the recognition of the paper’s relevance to robotics and visual RL, as well as the extensive experimental evaluation. The comments helped us identify several places where the method description could be made clearer. We have revised the manuscript and will address your questions below. The revision in the manuscript is highlighted in blue.
>
> > *It is unclear how the natural language prompt used by the language-conditioned segmenter is obtained. Is it manually written for each task and input, or generated automatically from the task description in some way? The prompt must be accurate in describing what is relevant to the task, so it deserves more explanation.*
>
> In our implementation, the prompt is fixed for each environment/task and is provided manually once before the mask-distillation phase. It is not changed per frame, per episode, or per seed, and it is not optimized using test performance. The prompt is a short task-level description that names the controllable agent component and the task-relevant object or objects. For example, in the Hammer task, the prompt specifies the robot hand, hammer, and nail block i.e. "Show the robot hand, Show the hammer, Show the nail block." This prompt is used only during the offline segmentation-supervision stage to obtain masks from SAMWISE. After this stage, the language-conditioned segmenter and the prompt interface are removed, and the RL agent is trained and evaluated using only the learned lightweight masker.
>
> We have revised the manuscript to make this explicit. Specifically, we added a prompt-protocol paragraph in Section 4.2.1, immediately after introducing the language prompt used by the segmenter. We also added the full list of task-level prompts in the Appendix A.1.2.
>
> > *The actor input is confusing as it includes representations of both the clean (unaugmented) and augmented images ($z_t$ and $\tilde{z}_t$). Why is it designed this way? At inference time, how is the clean representation available when the agent only observes inputs with noisy background?*
>
> We thank the reviewer for pointing this out. The original formulation could be misread as if the actor receives both the clean representation $z_t$ and the augmented representation $\tilde{z}_t$, as simultaneous inputs. This is not the intended design. Following SADA, we use $z_t$ and the augmented representation $\tilde{z}_t$, as simultaneous inputs. This is not the intended design.Following SADA, we use $z_t$ and $\tilde{z}_t$ as clean and augmented training streams respectively, that are concatenated along the batch dimension. Thus, $\pi([z_t,\tilde{z}_t]_N)$ means that the same single-input actor is applied batch-wise to clean and augmented masked representations. It does not define a two-input actor.
>
> During actor updates, the policy is exposed to both clean and augmented masked views, while the critic evaluates the corresponding actions through the clean stream. This follows the asymmetric update used in SADA [1] where the policy is regularized by augmented observations, while value estimation remains anchored to the lower-variance clean representation. At inference time, no clean counterpart is required or assumed to be available. The deployed agent observes only the current image, applies the trained student masker, encodes the resulting masked observation, and acts from this single representation.
>
> We revised Section 4.3 to make this explicit by replacing the two-input actor notation with the batch-wise SADA formulation. We also adapted Figure 2 and its caption accordingly.
>
> > *The asymmetric training strategy can be better explained. The sentence "During critic updates, the online critic is optimized on clean and augmented masked views, whereas the bootstrap target is computed from the clean view only." is somewhat confusing. Can the authors provide explicit equations for the training loss for the critic?*
>
> We thank the reviewer for noting that the critic update was described too compactly. We agree that the previous sentence did not make clear how the clean and augmented masked views enter the critic objective. We have revised Section 4.3 to explicitly define the clean masked representation $z_t$, the augmented masked representation $\tilde{z}\_{t}$, the bootstrap target, and the critic loss. In the revised formulation, we explain that the online critic is trained on both $z_t$ and $\tilde{z}\_{t}$, while the temporal-difference target is computed only from the clean next-state representation $z_{t+1}$. We explicitly define the target Q-value and the critic loss. This makes explicit that both the clean and augmented current views are regressed to the same clean-view bootstrap target. The clean target avoids introducing additional augmentation-induced variance into the temporal-difference estimate, while the augmented critic term still regularizes the critic on corrupted masked observations.

---

> ### Author Response · Authors · 2026-05-27
> **Response to Reviewer AcQJ-II**
>
> > *It is unclear why the student masker is trainable also during RL. The student masker has been trained in the Augmentation-Consistent Masker Phase. Keeping it trainable during RL introduces undesirable variation and instability. Why is this needed or preferred?*
>
> We thank the reviewer for raising this point. We agree that updating the student masker directly through actor-critic gradients could introduce undesirable mask drift and instability. This is not the intended role of the student masker. The student masker is not optimized by the RL objective. Instead, after the clean teacher masker $\psi_m$​ has been distilled from SAMWISE masks and frozen, the student masker $\psi_m^\star$​ is trained only through the augmentation-consistency objective against this fixed teacher.
>
> The reason for introducing the student is that the clean teacher $\psi_m$​ is trained only on unaugmented observations and therefore provides a stable semantic target, but it is not explicitly trained to handle augmented or visually corrupted inputs. The student receives augmented observations and learns to recover the corresponding clean teacher mask. This transfers the semantic prior from the clean teacher to an augmentation-robust masker, without requiring SAMWISE during RL training or inference. During RL, if the student is updated, it is updated only by the same teacher-student consistency loss on replay-buffer observations; gradients from the actor and critic are stopped at the masked observation. Thus, the semantic target remains fixed, and the masker is not allowed to drift according to sparse or noisy RL gradients.
>
> We revised Section 4.2.2 and Section 4.3 to clarify this distinction. In particular, we now state that $\psi_m$​ remains frozen, $\psi_m^\star$​ is updated only through the consistency loss against the frozen teacher, and actor-critic losses do not update either masker. At evaluation, the student masker is frozen and used as the deployed perceptual preprocessing module.
>
> We have also revised Figure 2 by adding a stop-gradient marker. We additionally rephrased the caption to make explicit that $\psi_m^\star$​, is trained only through $\mathcal{L}_{masker}$, against the frozen teacher.
>
> > *The need for two separate maskers is not fully justified. It is unclear why a single masker could not be trained on both unaugmented and augmented inputs. Can the authors compare with this simpler alternative or justify the original design?*
>
> We thank the reviewer for pointing out that the motivation for using two separate maskers was not sufficiently explicit. We have revised Section 4.2.2 to clarify that the two-masker design is intended to decouple semantic distillation from augmentation robustness.
> The teacher masker $\psi_m$, is trained only on clean, unaugmented observations to preserve a stable semantic target distilled from the language-conditioned segmenter. The student masker $\psi_m^\star$, is then trained on augmented observations to reproduce this fixed clean target. This separation is important because a single masker trained jointly on clean and augmented inputs must preserve clean semantic precision while also becoming invariant to strong visual corruption using the same parameters. Under strong overlay augmentations, these objectives can conflict: adapting to corrupted inputs may degrade the clean semantic anchor. Our design avoids this by keeping the teacher fixed and allowing only the student to absorb augmentation-induced variation.
>
> Thus, the teacher provides a stationary semantic reference, while the student learns the deployment-facing augmentation-robust mapping. At evaluation, only the trained student masker is deployed, so the two-masker design does not add inference-time segmentation overhead.
>
> To further address the reviewer’s concern (and as suggested by Reviewer cz8Y), we have added a conceptual justification in Section 4.2.2 and are additionally evaluating a single-masker ablation to compare against the simpler alternative.

---

> ### Author Response · Authors · 2026-05-27
> **Response to Reviewer AcQJ-III**
>
> **Additional Comments:**
> > *The augmentation setup uses only one overlay strength $(\delta=0.5)$. The authors should discuss whether the method could generalize to other values or report the results in those cases.*
>
> In our experiments, we use $(\delta=0.5)$ for the SVEA-style overlay augmentation, and we apply the same augmentation setting across all compared methods to keep the evaluation controlled and fair. TaLaS is not architecturally tied to  $(\delta=0.5)$. Smaller values correspond to stronger visual corruption, whereas larger values retain more of the original observation. We expect the learned student masker to tolerate moderate variation in this coefficient because it is trained to recover the clean teacher mask from corrupted observations. However, we do not claim invariance to arbitrary overlay strengths, and a full sensitivity analysis over $\delta$ is an additional robustness study. We now state this explicitly in the manuscript.
>
> > *Grammar: "our objective is to learn a policy $\pi$ consistent policy performance across this family of environments..."*
>
> We thank the reviewer for pointing this out. We have corrected the sentence in Section 4.1. The revised text now states "our objective is to learn a policy $\pi$ that maintains consistent performance".
>
> > *Grammar: "Both, however, utilizes segmentation model..., adding to substantial computational and overhead reliance on instance-level labels."*
>
> We thank the reviewer for identifying this grammatical issue. We have revised the sentence in Section 2.2. The revised version states "Both methods, however, use a segmentation model during training and inference, introducing substantial computational overhead and, in some cases, reliance on instance-level labels (Section 5.6.2)".
>
> > *POMDP should be defined.*
>
> We thank the reviewer for noting this omission. We have revised Section 3.1 to explicitly define POMDP as a *Partially Observable Markov Decision Process* before introducing its formal tuple.
>
> **References**\
> [1] Almuzairee, Abdulaziz, Nicklas Hansen, and Henrik I. Christensen. "A Recipe for Unbounded Data Augmentation in Visual Reinforcement Learning." Reinforcement Learning Conference.

---

> ### Author Response · Authors · 2026-06-06
> **Additional Response to Reviewer AcQJ**
>
> > *The need for two separate maskers is not fully justified. It is unclear why a single masker could not be trained on both unaugmented and augmented inputs. Can the authors compare with this simpler alternative or justify the original design?*
>
> To further address the reviewer’s concern, we have now added a single-masker ablation in Section 5.6.4. This ablation compares full TaLaS against a simpler variant in which one masker is trained directly to predict SAMWISE masks without the teacher-student consistency stage. We evaluate this comparison on Walker Walk over five random seeds. The single-stage variant obtains $842 \pm 61$ in video-easy and $664 \pm 97$ in video-hard, whereas full TaLaS achieves $878 \pm 33$ and $787 \pm 50$, respectively. These results show that the proposed two-stage design improves performance, especially under stronger visual distraction, and supports our choice of separating semantic distillation from augmentation-consistent student training.

---

> > ### Comment · Reviewer_AcQJ · 2026-06-13
> >
> > Thank the authors for their clarification. The responses, and the additional experiment on the necessity of separately distilling the language-conditioned segmentor and training the augmentation-robust student, are very informative.

---

### Review · Reviewer_cz8Y · 2026-05-25

**Summary Of Contributions:**

The paper introduces **TaLaS** (Task-Relevant Language-conditioned Segmentation), a framework for improving zero-shot generalization in visual reinforcement learning under distribution shift (distractors, texture/lighting changes, background video). The central idea is to use a language-conditioned segmentation foundation model (SAMWISE, built on SAM2) only *once*, offline, to provide semantic supervision, and to distill its task-relevant masks into a lightweight CNN masker that runs cheaply at training and inference time. The pipeline has two phases:

1. **Phase I (language-guided distillation):** A frozen language-conditioned segmenter produces task masks on clean observations collected under a random exploration policy; a compact CNN masker is trained via BCE to imitate these masks.
2. **Phase II (augmentation-consistent student):** The Phase-I masker is frozen as a teacher, and a student masker is trained on augmented observations via consistency regularization to reproduce the teacher's clean-input masks, yielding augmentation-stable masks.

These masks gate observations (Hadamard product) before a frozen pretrained encoder (PIE-G / DrQ-v2 backbone). To handle the mask distribution shift at deployment, the authors add a SADA-inspired **asymmetric actor–critic update**: the actor sees clean and augmented masked views while the critic/bootstrap target uses clean views only. Evaluation is on RL-ViGen across DMC-GB, Unitree locomotion, and Adroit manipulation, in video-easy and video-hard settings, against ten visual-RL baselines.

**Key strengths:**
- The "use the expensive segmenter once, distill into a cheap masker" framing is a clean and practically valuable idea. The wall-clock comparison (≈10 hours vs. an estimated ≈3 days for online SAMWISE) makes the efficiency motivation concrete and is one of the paper's strongest contributions.
- The video-hard results are the most convincing evidence: a reported 13.4% gain over the next-best method on DMC-GB video-hard, and a small easy→hard degradation (7.5%) versus 35%/19% for SRM/PIEG, tell a coherent story that semantic masking matters most precisely when distraction is severe.
- The asymmetric-training ablation (Fig. 5) and the t-SNE clustering analysis (Fig. 4) support the central claims reasonably well.

**Key weaknesses:**
- The empirical advantage is uneven and is framed more favorably than the tables support, especially in easy settings and on some specific tasks.
- The closest competitors conceptually (segmentation-based maskers FTD and SAM-G) are discussed at length but never empirically compared against. Baseline coverage is also inconsistent across tables.
- Statistical support is thin for the environments carrying the largest claimed gains (3 seeds, large variance, no effect-size/aggregate statistics).
- Several ablations needed to attribute the gains to the proposed components (distillation vs. consistency vs. prompt) are missing.
- Notational inconsistencies and writing errors hurt clarity.

**Audience:**

Yes

**Audience Explanation:**

Visual-RL generalization is an active subfield, and the specific question of how to obtain the semantic benefits of foundation-model segmentation without paying its inference cost is of clear interest. The offline-distillation framing, the wall-time analysis, and the robustness results under heavy distraction would be useful to researchers working on representation learning, masking/saliency methods, and sim-to-real transfer.

**Broader Impact Concerns:**

I have no significant ethical concerns specific to this work; it studies generalization on standard simulated control benchmarks. No Broader Impact Statement is strictly required, though a brief note on failure modes relevant to real-world deployment (e.g., the acknowledged risk that segmentation-based filtering degrades when the object of interest itself changes appearance) would be a reasonable addition given the paper's stated deployment motivation.

**Claims And Evidence:**

No

**Claims Explanation:**

The *efficiency* claim (avoiding online segmentation) is well supported. The *robustness-under-heavy-distraction* claim is plausibly supported on DMC-GB video-hard. However, several headline claims are not yet adequately supported:

1. **Inconsistency between framing and results.** The abstract and introduction present TaLaS as broadly superior, but the method is *not* best on average in DMC video-easy (Avg 716 vs. SRM 763, SVEA 757), is near-worst on Unitree Stand video-easy (325, below DrQ's 341), and underperforms on Cheetah Run in *both* easy and hard settings (e.g., 219 vs. CNSN 347 in VE). The method is best characterized as specialized for high-distraction regimes; the current text obscures real losses.

2. **Missing comparison to the most direct competitors.** FTD (Chen et al., 2024) and SAM-G (Wang et al., 2023) are repeatedly positioned as the work TaLaS improves upon (segmentation-based masking), yet neither appears in any results table. The efficiency advantage over these methods is argued, but their *performance* relative to TaLaS is never shown. Additionally, MaDi—a direct masking baseline—appears in Table 1 (DMC) but is dropped from Table 2 (Unitree) and Fig. 3 (Adroit), where the largest percentage gains are claimed.

3. **Weak statistical support where gains are largest.** Adroit and Unitree results use only 3 seeds, with large reported standard deviations (e.g., WS VH 824±287, FS VH 556±291). Many of the claimed gaps (e.g., "48% over PIE-G" on Adroit VE) are likely within noise. No confidence intervals, significance tests, or aggregate metrics (e.g., IQM / stratified bootstrap, per Agarwal et al., 2021) are reported, which is now standard practice for generalization claims in visual RL.

4. **Component attribution is incomplete.** The only ablation isolates the asymmetric actor–critic update. There is no ablation isolating the contribution of Phase II consistency vs. Phase I distillation alone, nor any sensitivity analysis on the language prompt—despite the limitations section acknowledging that mask quality depends heavily on prompt quality. As written, the reader cannot determine which components drive the gains.

I would be glad to revise this assessment to "Yes" if the requested experiments and reframing below are addressed; the core idea is sound and the video-hard evidence is promising.

**Requested Changes:**

1. **Add a direct comparison to FTD and/or SAM-G** on at least one shared benchmark (ideally DMC-GB video-hard), including both performance and wall time. Without this, the central positioning of the paper is unsupported.
2. **Report aggregate statistics and effect sizes.** Add IQM with stratified bootstrap confidence intervals (or at minimum per-task confidence intervals and significance tests) for the headline claims, especially on Adroit/Unitree. Increase the seed count on Adroit/Unitree from 3 to match the 5 used for DMC, or justify the discrepancy.
3. **Add ablations attributing the gains to specific components:** (a) Phase-I masker alone (no consistency phase) vs. full pipeline; (b) sensitivity to the language prompt (e.g., a vague/under-specified prompt and an over-specified prompt).
4. **Reframe the claims to match the evidence.** Explicitly state that TaLaS targets high-distraction regimes, acknowledge the video-easy and Cheetah Run results, and soften the "outperforms across the board" framing in the abstract/intro.
5. Make baseline coverage consistent across Tables 1–2 and Fig. 3 (include MaDi everywhere, or explain its omission).
6. Analyze the Cheetah Run failure mode—does the masker drop motion-relevant pixels? A qualitative mask visualization on this task would help.
7. Fix notational inconsistencies: the student masker is written as both ψ⋆_m and ψ_s; the masked augmented input uses ψ_m(õ_t) in one place and ψ⋆_m elsewhere; clarify how the actor combines z_t and z̃_t and what it conditions on at inference when only clean masks exist.
8. Reconcile Figure 1 (labeled "SAM") with the text (SAMWISE); the text-conditioned vs. point-prompted distinction is material to the contribution.
9. Proofread for grammar/typos, e.g., "modular and and self-supervised" (p.6, repeated word) and the broken sentence on p.5 ("...learn a policy π consistent policy performance across this family...").
10. Quantify the "10 strong baselines" (p.2) claim more carefully, since several baselines appear in only one of the three benchmark settings.

---

> ### Author Response · Authors · 2026-06-08
> **Response to Reviewer cz8Y**
>
> We thank the reviewer for the detailed and constructive assessment. We appreciate the recognition that the offline distillation of language-conditioned segmentation into a lightweight masker is practically valuable, and that the video-hard results and wall-clock analysis support the motivation of TaLaS. The reviewer’s comments helped us better align the paper’s claims with the evidence, strengthen the empirical analysis, and clarify several methodological details. We have revised the manuscript accordingly. The revision in the manuscript is highlighted in blue.
>
> > *Add a direct comparison to FTD and/or SAM-G on at least one shared benchmark (ideally DMC-GB video-hard), including both performance and wall time. Without this, the central positioning of the paper is unsupported.*
>
> We thank the reviewer for this suggestion. We agree that FTD is one of the closest segmentation-assisted RL baselines. We have therefore added a comparison with FTD in Appendix A.5 and Table 6. Specifically, we compare against the reported FTD results on the DMC tasks common to both papers: Cartpole Swingup, Finger Spin, Cheetah Run, and Walker Walk. These results are directly obtained from the paper itself [1]. To further extend the comparison within our RL-ViGen setting, we additionally run FTD on Walker Stand over three random seeds.
>
> TaLaS improves over FTD on Cartpole Swingup, Finger Spin, Walker Walk, and Walker Stand, while FTD performs better on Cheetah Run. Due to the high cost of online segmentation, running a full five-seed FTD reproduction across all DMC tasks was not feasible within the rebuttal period. We therefore include the shared-task comparison and retain the wall-time analysis in Section 5.6.1 to highlight the central computational distinction: FTD relies on online segmentation and attention, whereas TaLaS distills the segmentation prior into a lightweight masker used during training and evaluation.
>
>
> > *Report aggregate statistics and effect sizes. Add IQM with stratified bootstrap confidence intervals (or at minimum per-task confidence intervals and significance tests) for the headline claims, especially on Adroit/Unitree. Increase the seed count on Adroit/Unitree from 3 to match the 5 used for DMC, or justify the discrepancy.*
>
> We thank the reviewer for raising this important point. We agree that visual RL can exhibit substantial seed-level variance and that aggregate statistics make the empirical comparison more reliable.
>
> To address this, we have updated the Adroit manipulation experiments from three to five random seeds and revised Section 5.5 and Figure 4 accordingly. We now report mean and standard deviation over five seeds for Pen, Door, and Hammer. In addition, we added Appendix A.6, where we report aggregate interquartile mean (IQM) scores with (95%) stratified bootstrap confidence intervals. The IQM scores are computed over task-normalized returns pooled across the Adroit tasks, and the corresponding aggregate values are reported in Table 7 and visualized in Figure 9.
>
> For Unitree, we retain the three-seed evaluation because quadruped training is substantially more computationally expensive, making a full five-seed rerun infeasible within the rebuttal window.

---

> ### Author Response · Authors · 2026-06-08
> **Response to Reviewer cz8Y-II**
>
> > *Add ablations attributing the gains to specific components: (a) Phase-I masker alone (no consistency phase) vs. full pipeline; (b) sensitivity to the language prompt (e.g., a vague/under-specified prompt and an over-specified prompt).*
> We thank the reviewer for this suggestion. We have added new component-level ablations to better isolate the contribution of the masker design and the prompt choice.
>
> First, we added Section 5.6.4, titled “Effect of Two-Stage Masker Distillation,” where we compare full TaLaS against a single-stage masker variant on Walker Walk over five random seeds. In this variant, the masker is trained directly to predict SAMWISE masks without the teacher-student consistency stage. The single-stage variant obtains $842 \pm 61$ in video-easy and $664 \pm 97$ in video-hard, whereas full TaLaS achieves $878 \pm 33$ and $787 \pm 50$, respectively. This corresponds to a $4.3$\% improvement in video-easy and an $18.6$\% improvement in video-hard, showing that the two-stage design is particularly beneficial under stronger visual distraction.
>
> Second, we added a prompt-sensitivity analysis in Section 5.6.3 and Appendix A.4.1. For each task, we manually annotate task-relevant regions on 10 random frames and compute IoU between the human annotation and masks produced by the default task-level prompt or a weaker coarse prompt. The default prompts achieve higher average IoU than the coarse prompts, indicating that prompt specificity improves the quality of the segmentation supervision.
>
> Together, these additions isolate the effects of both the two-stage masker distillation and the task-level prompt design, addressing the reviewer’s request for more detailed component analysis.
>
>
> > *Reframe the claims to match the evidence. Explicitly state that TaLaS targets high-distraction regimes, acknowledge the video-easy and Cheetah Run results, and soften the "outperforms across the board" framing in the abstract/intro.*
>
> We thank the reviewer for this important point. We agree that the claims should more precisely reflect the empirical evidence. We have therefore revised the framing in three places: the abstract, the final paragraph of the introduction, and the discussion of DMC results in Section 5.3.1.
>
> In the abstract and introduction, we now state that TaLaS achieves its strongest gains under challenging video-hard visual shifts, where semantic distractor suppression is most critical, while remaining competitive in easier settings. This better reflects the intended regime of the method without implying uniform dominance across all visual conditions.
>
> In Section 5.3.1, we also explicitly discuss the task-level variation in the results. In particular, we note that TaLaS achieves the best average performance in the video-hard setting, but the improvements are not uniform across all tasks. We specifically acknowledge Cheetah Run as an exception where other methods obtain higher returns. This revised discussion clarifies that the benefit of semantic masking is task-dependent, while the aggregate video-hard results show that TaLaS is most effective when suppressing visual distractors is central to generalization.
>
>
> > *Make baseline coverage consistent across Tables 1–2 and Fig. 3 (include MaDi everywhere, or explain its omission).*
>
> We thank the reviewer for pointing out the difference in baseline coverage across benchmarks. We have clarified this in Section 5.1. For MaDi, we use the results reported in the original paper for the corresponding DMC-GB setting, which is why MaDi appears in Table 1. However, MaDi does not report results for the Unitree and Adroit settings considered in our evaluation. Therefore, we do not include it in Table 2 or Figure 3. To keep the comparisons fair and avoid introducing non-comparable reimplementations, we include baselines only when official or previously reported results are available under the same benchmark protocol.
>
>
> > *Analyze the Cheetah Run failure mode—does the masker drop motion-relevant pixels? A qualitative mask visualization on this task would help.*
>
> We thank the reviewer for pointing out this failure case. We agree that Cheetah Run is the clearest task where TaLaS does not outperform the strongest baselines.
>
> We have added a qualitative mask visualization in Figure 8 (bottom row) and explicitly discuss this failure mode in Section 5.3.1. Figure 8 includes the original video-hard observation and the corresponding masked observation for Cheetah Run, alongside other representative tasks. The visualization suggests that, under strong video-hard distractors, the masker can become conservative around thin, low-contrast limb regions. Since Cheetah Run depends on fast forward locomotion, these limb pixels carry motion-relevant information; partially attenuating them can therefore hurt control performance. This explains why TaLaS is less competitive on Cheetah Run, while remaining effective on most other distractor-heavy tasks.

---

> ### Author Response · Authors · 2026-06-08
> **Response to Reviewer cz8Y-III**
>
> > *Fix notational inconsistencies: the student masker is written as both ψ⋆_m and ψ_s; the masked augmented input uses ψ_m(õ_t) in one place and ψ⋆_m elsewhere; clarify how the actor combines z_t and z̃_t and what it conditions on at inference when only clean masks exist.*
>
> We thank the reviewer for pointing out these notational and conceptual inconsistencies. We have revised Section 4.2.2, Section 4.3, and Figure 2 to make the notation and data flow consistent.
>
> First, we now consistently denote the Phase-II student masker as $\psi_m^\star$ and use it for augmented observations.
> In Section 4.3, the augmented masked observation is explicitly defined as $\tilde{o}_t \odot \psi_m^\star(\tilde{o}_t)$, while the clean masked observation is produced by the frozen teacher masker $\psi_m$. Second, we clarified that the actor does not receive $z_t$ and $\tilde{z}_t$ as simultaneous feature inputs. Instead, $[z_t,\tilde{z}_t]_N$ denotes concatenation along the batch dimension, so the same single-input actor is applied batch-wise to clean and augmented masked representations. Third, we added the inference-time policy path to make clear that no clean counterpart is required at deployment: the deployed agent uses only the current observation, the trained student masker, and the encoder before applying the actor.
>
> We also updated Figure 2 and its caption to show the batch-wise concatenation with a red rectangle and the stop-gradient marker, clarifying that actor-critic gradients do not update $\psi_m^\star$.
>
>
> > *Reconcile Figure 1 (labeled "SAM") with the text (SAMWISE); the text-conditioned vs. point-prompted distinction is material to the contribution.*
>
> We thank the reviewer for pointing out this inconsistency. We have revised Figure 1 and its caption to replace the generic “SAM” label with “SAMWISE,” matching the method description in the text.
>
>
> > *Proofread for grammar/typos, e.g., "modular and and self-supervised" (p.6, repeated word) and the broken sentence on p.5 ("...learn a policy π consistent policy performance across this family...").*
>
> We thank the reviewer for pointing out these grammar and typographical issues. We have proofread the manuscript and corrected the highlighted errors, including the repeated-word issue, the broken objective sentence in Section 4.1, and the segmentation-baseline sentence in Section 2.2. We also performed an additional pass to correct remaining grammar issues throughout the manuscript, including phrasing in the Introduction, Section 3.1 and Section 5.3.3.
>
>
> > *Quantify the "10 strong baselines" (p.2) claim more carefully, since several baselines appear in only one of the three benchmark settings.*
>
> We thank the reviewer for pointing this out. We have revised the manuscript to make the scope of our claims more precise. Specifically, we changed the final sentence of the Introduction to state that TaLaS shows its strongest benefits under challenging distractor-heavy settings, while remaining competitive in easier evaluation regimes. This avoids implying uniform superiority across all settings.
>
> We also revised Section 5.1, “Baseline Methods,” to clarify the baseline coverage. The manuscript now states that we compare against up to ten visual RL baselines across benchmarks, with the exact set depending on implementation availability, protocol compatibility, and reported results. This avoids implying that every baseline is evaluated in every benchmark setting.
>
>
> **References**
>
> [1] Chao Chen, Jiacheng Xu, Weijian Liao, Hao Ding, Zongzhang Zhang, Yang Yu, and Rui Zhao. Focus-then-decide: segmentation-assisted reinforcement learning. In Proc. of the AAAI Conf. on Artificial Intelligence,volume 38, 2024.

---

> > ### Comment · Reviewer_cz8Y · 2026-06-15
> >
> > I thank the authors for the detailed rebuttal and revised manuscript. I have read the response carefully and appreciate the additional experiments, clarified framing, and corrections made in the revision. The changes address many of my main concerns, while noting that some limitations such as partial baseline coverage and remaining seed constraints should remain clearly stated in the final version.

---

### Author Response · Authors · 2026-06-08
**Rebuttal Summary**

We thank all reviewers for their time to evaluate our work. We are pleased that all reviewers found our contribution to be valuable, significant and our results to be convincing.

**cz8Y:** "clean and practically valuable idea", "concrete motivation for efficiency", "video-hard results are the most convincing evidence" and "asymmetric-training ablation and t-SNE clustering support the central claims".\
**AcQJ:** "more efficient than using language-conditioned segmenters directly",  "good performance" and "benefit in computational efficiency".\
**83Au:** "TaLaS often outperforms strong baselines" and "less sensitive to background changes".

In response to the reviewers' feedback during the rebuttal phase, we have addressed their concerns and made following revisions to the manuscript:
* Added comparison with FTD, included IQM, added masker ablation, added more seeds, soften claims and corrected grammar, as suggested by **reviewer cz8Y**.

* Added natural language prompt analysis and added explanation in the main text for clarification, as suggested by **reviewer AcQJ**.

* Added masked ablation study, added natural language prompt analysis, added qualitative analysis of video-hard tasks, as suggested by **reviewer 83Au**.

We uploaded a revised version of the main paper and the supplementary material, with all changes marked in blue color.

We have also provided detailed responses to each of the reviewers' comments below.

---

### Decision · Action_Editor_BN8F · 2026-07-03

**Recommendation:** Accept with minor revision

**Additional Comments:**

TaLaS addresses zero-shot generalization in visual reinforcement learning by distilling the task-relevant masks of an expensive language-conditioned segmentation foundation model, used only once offline, into a lightweight masker that is stabilized through a two-stage teacher-student consistency scheme and paired with an asymmetric actor-critic update so that no online segmentation is required at deployment. The reviewers consistently found the core idea clean and practically valuable, praised the wall-clock efficiency analysis for making the motivation concrete, and agreed that the approach is most compelling under heavy visual distraction, where the video-hard results together with the t-SNE and ablation analyses tell a coherent story, and all reviewers regarded the findings as relevant to the TMLR audience. The principal concerns were that the empirical advantage is uneven and was initially framed more favorably than the tables supported, that the closest segmentation-based methods such as FTD and SAM-G were discussed but not compared against, that statistical support was thin on the manipulation and locomotion benchmarks, that several component and prompt-sensitivity ablations were missing, and that the contribution reads as an effective integration of existing components rather than a fundamentally new algorithmic advance. During the rebuttal the authors responded substantively by adding an FTD comparison, reporting interquartile-mean scores with stratified bootstrap confidence intervals and increasing the number of seeds on the manipulation benchmark, introducing a two-stage masker ablation and a prompt-sensitivity study, analyzing the Cheetah Run failure mode with mask visualizations, reframing the abstract and introduction to emphasize the high-distraction regime, and correcting the notational and grammatical issues, which prompted one reviewer to raise their recommendation and left the others positive while a residual reservation about the strength of the evidence for a few headline claims and the modest novelty remained. A small number of clarifications and further tightening of the remaining claims would strengthen the final version.

**Audience:**

Yes

**Audience Explanation:**

Obtaining the semantic benefits of foundation-model segmentation without paying its inference cost is a question of clear interest to researchers working on visual reinforcement learning, representation learning, and sim-to-real transfer. The offline-distillation framing, the efficiency analysis, and the robustness results under strong visual shifts are useful to this community, and all reviewers agreed that the findings would interest the TMLR audience.

**Claims And Evidence:**

Yes

**Claims Explanation:**

The efficiency claim of avoiding online segmentation is well supported by the wall-clock analysis, and the robustness gains under heavy distraction are convincingly demonstrated on the video-hard benchmark. Following the rebuttal, the authors added a comparison to a segmentation-based baseline, reported aggregate interquartile-mean scores with stratified bootstrap confidence intervals, increased the number of seeds, added component-level and prompt-sensitivity ablations, and reframed the claims to match the evidence, so the central claims are now adequately supported. A few headline claims still rest on relatively limited statistical support, which motivates minor further tightening.